# Function of hTim8a in complex IV assembly in neuronal cells provides insight into pathomechanism underlying Mohr-Tranebjærg syndrome

Yilin Kang[1,2], Alexander J Anderson[1,2], Thomas Daniel Jackson[1,2], Catherine S Palmer[1,2], David P De Souza[3], Kenji M Fujihara[4,5], Tegan Stait[6,7], Ann E Frazier[6,7], Nicholas J Clemons[4,5], Deidreia Tull[3], David R Thorburn[6,7,8], Malcolm J McConville[3], Michael T Ryan[9], David A Stroud[1,2], Diana Stojanovski[1,2]*

[1]Department of Biochemistry and Molecular Biology, The University of Melbourne, Melbourne, Australia; [2]The Bio21 Molecular Science and Biotechnology Institute, The University of Melbourne, Melbourne, Australia; [3]Metabolomics Australia, The Bio21 Molecular Science and Biotechnology Institute, The University of Melbourne, Melbourne, Australia; [4]Division of Cancer Research, Peter MacCallum Cancer Centre, Melbourne, Australia; [5]Sir Peter MacCallum Department of Oncology, The University of Melbourne, Melbourne, Australia; [6]Murdoch Children's Research Institute, Royal Children's Hospital, Melbourne, Australia; [7]Department of Paediatrics, University of Melbourne, Melbourne, Australia; [8]Victorian Clinical Genetic Services, Royal Children's Hospital, Melbourne, Australia; [9]Department of Biochemistry and Molecular Biology, Monash Biomedicine Discovery Institute, Monash University, Melbourne, Australia

**Abstract** Human Tim8a and Tim8b are members of an intermembrane space chaperone network, known as the small TIM family. Mutations in *TIMM8A* cause a neurodegenerative disease, Mohr-Tranebjærg syndrome (MTS), which is characterised by sensorineural hearing loss, dystonia and blindness. Nothing is known about the function of hTim8a in neuronal cells or how mutation of this protein leads to a neurodegenerative disease. We show that hTim8a is required for the assembly of Complex IV in neurons, which is mediated through a transient interaction with Complex IV assembly factors, in particular the copper chaperone COX17. Complex IV assembly defects resulting from loss of hTim8a leads to oxidative stress and changes to key apoptotic regulators, including cytochrome c, which primes cells for death. Alleviation of oxidative stress with Vitamin E treatment rescues cells from apoptotic vulnerability. We hypothesise that enhanced sensitivity of neuronal cells to apoptosis is the underlying mechanism of MTS.

*For correspondence:
d.stojanovski@unimelb.edu.au

**Competing interests:** The authors declare that no competing interests exist.

## Introduction

Mitochondria are fundamental cellular organelles governing many metabolic processes including fatty acid oxidation, the Krebs cycle, oxidative phosphorylation (OXPHOS), and fatty acid ß-oxidation (*Kasahara and Scorrano, 2014*; *McBride et al., 2006*). Organelle dysfunction is associated with a broad spectrum of diseases, including mitochondrial diseases that are genetic, often inherited disorders associated with energy generation defects (*Nunnari and Suomalainen, 2012*; *Frazier et al., 2019*). The health and functionality of the mitochondrial network relies on the biogenesis of approximately 1500 nuclear encoded proteins, which are imported into mitochondria using sophisticated

translocation machines (*Chacinska et al., 2009*; *Endo and Yamano, 2009*). The biogenesis of many hydrophobic membrane proteins relies on the small TIM proteins, a chaperone network in the intermembrane space that function to shield hydrophobic proteins as they passage this aqueous space (*Curran et al., 2002a*; *Curran et al., 2002b*; *Endres et al., 1999*; *Hoppins and Nargang, 2004*; *Wiedemann et al., 2004*).

Unlike yeast that have five well-characterised small TIM proteins, Tim8, Tim9, Tim10, Tim12 and Tim13 (*Koehler, 2004*; *Koehler et al., 1999*; *Koehler et al., 1998*), human cells have six small TIM proteins that have been poorly characterised, these include, hTim8a, hTim8b, hTim9, hTim10a, hTim10b, and hTim13 (*Bauer et al., 1999*; *Gentle et al., 2007*). Hexameric assemblies consisting of yeast Tim8-Tim13 and Tim9-Tim10 function to transfer hydrophobic proteins through the intermembrane space to inner and outer membrane translocases (*Hoppins and Nargang, 2004*; *Kang et al., 2016*; *Kang et al., 2017*; *Wiedemann et al., 2004*). The yeast Tim8-Tim13 complex is believed to be responsible for the delivery of TIM substrates (Tim17, Tim22 and Tim23) to the TIM22 complex (*Beverly et al., 2008*; *Curran et al., 2002b*; *Davis et al., 2007*; *Paschen et al., 2000*). However, the presence of hTim13 in protein interaction networks like the SPY complex, an inner membrane proteolytic hub (*Wai et al., 2016*), suggests that the small TIM proteins may have acquired novel functions in human mitochondria. Additionally, why human mitochondria have two Tim8 isoforms is unclear and the predominate expression of hTim8a in brain (*Jin et al., 1999*), suggests cell specific functions. In line with this, mutations in the *TIMM8A* gene that encodes the hTim8a protein, cause Mohr-Tranebjærg syndrome (MTS), an X-linked recessive neurodegenerative disorder characterised by progressive sensorineural hearing loss, dystonia, cortical blindness and dysphagia (*Jin et al., 1996*; *Koehler et al., 1999*; *Tranebjaerg et al., 1995*). Given the function of yeast Tim8 in the import of Tim23, it has been assumed that defects in the import of human Tim23 were the underlying basis of MTS (*Leuenberger et al., 1999*; *Paschen et al., 2000*; *Rothbauer et al., 2001*).

Using cell knock-out studies in HEK293 and the neuroblastoma cell line, SH-SY5Y, we uncover a novel function for hTim8a and hTim8b in the assembly of Complex IV (cytochrome *c* oxidase) in a cell-specific manner. Our data suggests that hTim8a function is more prominent in neuronal-like SH-SY5Y cells, while hTim8b function is more prominent in HEK293 cells. Consequently, depletion of hTim8a has a drastic impact on cell health in SH-SY5Y cells, with major impact to cell viability, mitochondrial membrane potential, perturbed Complex IV activity and oxidative stress. This cellular dysfunction is associated with changes to key apoptotic regulators, in particular cytochrome *c* that sensitises cells lacking hTim8a to intrinsic cell death. Alleviation of oxidative stress in cells lacking hTim8a by treatment with Vitamin E rescues cells from their apoptotic vulnerability and provides a molecular explanation for previously reported neuronal cell loss in MTS patients (*Tranebjaerg et al., 2001*). We suggest that early intervention with antioxidant could represent a treatment strategy for mitochondrial neuropathologies like Mohr-Tranebjærg syndrome.

## Results

### Loss of functional hTim8a or hTim8b reveals a role in Complex IV biogenesis

We set out to establish the function of hTim8a and hTim8b in human cells by targeting the genes using CRISPR/Cas9 in two cell models: (i) the widely used HEK293 cell line; and (ii) the neuroblastoma cell line SH-SY5Y, which we used as an in vitro model of neuronal function. We also targeted *TIMM9* in HEK293 cells as a control. *TIMM9* edited cells had two indel variants causing frame-shift mutations and new stop codons at 2 or four aa beyond the wildtype stop codon (*Figure 1—figure supplement 1A*), giving rise a slower migrating hTim9 mutant protein that was reduced at the steady-state level (*Figure 1—figure supplement 2A*, left panel). Given this, we refer to this cell line as hTim9$^{MUT}$ (MUT, mutant). HEK293 cells edited for *TIMM8A* resulted in a complete loss of the hTim8a protein and we refer to this cell line as and hTim8a$^{KO}$ (KO, knockout) (*Figure 1—figure supplement 2A*, middle panel). SH-SY5Y cells targeted for *TIMM8A* were heterozygous (contained a wild-type and modified allele) (*Figure 1—figure supplement 1C*), however isolated mitochondria had no hTim8a visible by western blot (*Figure 1—figure supplement 2A*, right panel) or via mass spectrometric analyses (Figure 2C). Given that the expression of hTim8a can be altered by skewed X-chromosome inactivation (*Plenge et al., 1999*), we hypothesise that the observed wild-type allele

of *TIMM8A* is located on an inactive X-chromosome and therefore refer to this cell line as hTim8a$^{MUT}$ $^{SH}$ ($^{SH}$ indicates SH-SY5Y). We also obtained knock-outs of *TIMM8B* in both HEK293 and SH-SY5Y cells (*Figure 1—figure supplement 1D and E*; *Figure 1—figure supplement 2A*) and refer to these cell lines as Tim8b$^{KO}$ and Tim8b$^{KO SH}$ ($^{SH}$ indicates SH-SY5Y).

We addressed the implications of depleting hTim9, hTim8a and hTim8b on the TIM22 complex and substrates of this inner membrane translocase in both HEK293 and SH-SY5Y cells. Mitochondria isolated from hTim9$^{MUT}$ cells show reduced steady state levels of TIM22 complex subunits (hTim22 and Tim29) and substrates of the TIM22 complex (ANT3, GC1 and hTim23) (*Figure 1—figure supplement 2A*), and displayed severe assembly defects of the TIM22 complex (*Figure 1A*, compare lanes 3 and 4). To the contrary, lack of functional hTim8a in both HEK293 and SH-SY5Y cells had no obvious impact on the TIM22 complex or TIM22 substrates when analysed by SDS-PAGE (*Figure 1—figure supplement 2A*) or BN-PAGE (*Figure 1B and C*; quantified for hTim8a$^{MUT SH}$ in *Figure 1—figure supplement 2B*). Likewise, depletion of hTim8b in HEK293 or SH-SY5Y cells had minimal impact on the levels of the TIM22 or TIM23 complex (*Figure 1D and E*; *Figure 1—figure supplement 2A*). Further analysis using in vitro mitochondrial import assays of [$^{35}$S]-hTim23, [$^{35}$S]-hTim22 and the glutamate carrier ([$^{35}$S]-GC1) into mitochondria isolated from hTim8a$^{KO}$ and hTim8b$^{KO}$ (*Figure 1F–H*) or hTim8a$^{MUT SH}$ cells (*Figure 1I*) suggests no defects in the assembly of these putative substrates. This data demonstrates that hTim9 is firmly embedded in the TIM22 complex biogenesis pathway, like its yeast counterpart, while hTim8a and hTim8b do not directly influence the TIM22 complex or pathway and may have alternative function(s) within human mitochondria. The normal import and assembly of hTim23 in the absence of hTim8a and hTim8b suggests there is an alternative pathomechanism underlying MTS.

To gain perspective into the potential function of hTim8a and hTim8b in human cells we performed label-free quantitative mass spectrometry on mitochondria isolated from hTim9$^{MUT}$, hTim8a$^{KO}$ (HEK293), hTim8a$^{MUT SH}$ (SH-SY5Y), hTim8b$^{KO}$ (HEK293) and hTim8b$^{KO SH}$ (SH-SY5Y) (*Figure 2A–2E*). hTim9$^{MUT}$ mitochondria showed reduced levels of hTim9 itself and its partner protein hTim10a, as the small TIM proteins are stable in their hexameric assemblies and not as monomeric proteins (*Baker et al., 2012*). In addition, the TIM22 complex subunits, Tim29 and hTim10b, and numerous mitochondrial carrier proteins (SLC25 family), including ANT1 (*SLC25A4*) and the Phosphate Carrier (*SLC25A3*) were significantly reduced in hTim9$^{MUT}$ (*Figure 2A* and *Figure 2—source data 1*). hTim8a$^{KO}$ mitochondria showed a decrease in the levels of: (i) hTim13 and hTim8b; (ii) the Apoptosis Inducing Factor, AIF; (iii) the Complex IV assembly factor, COX17; and (iv) two proteins belonging to the SLC25 family, SLC25A5 (ANT2) and SLC25A14, a previously described mitochondrial uncoupling protein (UCP5) (*Kim-Han et al., 2001*; *Sanchis et al., 1998*; *Figure 2B* and *Figure 2—source data 1*). hTim8a$^{KO}$ cells also displayed upregulated levels of the pro-apoptotic BCL-2 family member Bax. The absence of hTim8a in SH-SY5Y cells (*Figure 2C* and *Figure 2—source data 1*), showed a strikingly different profile to the HEK293 cells with changes to both mitochondrial and nuclear encoded Complex IV subunits and assembly factors. Specifically, reduced levels of mitochondrially-encoded COX2 (*MT-CO2*) and COX3 (*MT-CO3*) and nuclear-encoded COX6A1, COX6C and COX7A2 were observed, while COX17 was down-regulated and two putative assembly factors COA5 and COX15, were upregulated. This data pointed to a novel and likely cell-type specific function of hTim8a in Complex IV biology and we wondered if the hTim8b isoform was performing a similar role in HEK293 cells. Indeed, hTim8b$^{KO}$ HEK293 cells showed reduced levels in Complex IV assembly factors including COX17, COA4, COA5, COA7 and CMC4 (*Figure 2D* and *Figure 2—source data 1*), while depletion of hTim8b in SH-SY5Y cells also led to changes to both mitochondrial and nuclear encoded Complex IV subunits and assembly factors (*Figure 2E* and *Figure 2—source data 1*). Taken together this data suggest a role of hTim8a and hTim8b in the biogenesis of Complex IV.

## Loss of hTim8a sensitises cells to intrinsic cell death

A change that caught our attention in the proteomics data was the up and down-regulation of key apoptotic regulators in HEK293 cells lacking hTim8a, including down-regulation of the Apoptosis Inducing Factor (AIF) and upregulation in the pro-apoptotic Bax. Changes to AIF and Bax were not evident in hTim9$^{MUT}$ or hTim8b$^{KO}$ mitochondria, suggesting a specific cellular response due to the lack of hTim8a and a potential increased susceptibility to apoptosis. We confirmed changes to the levels of AIF and Bax by western blot and also uncovered an upregulation in the levels of

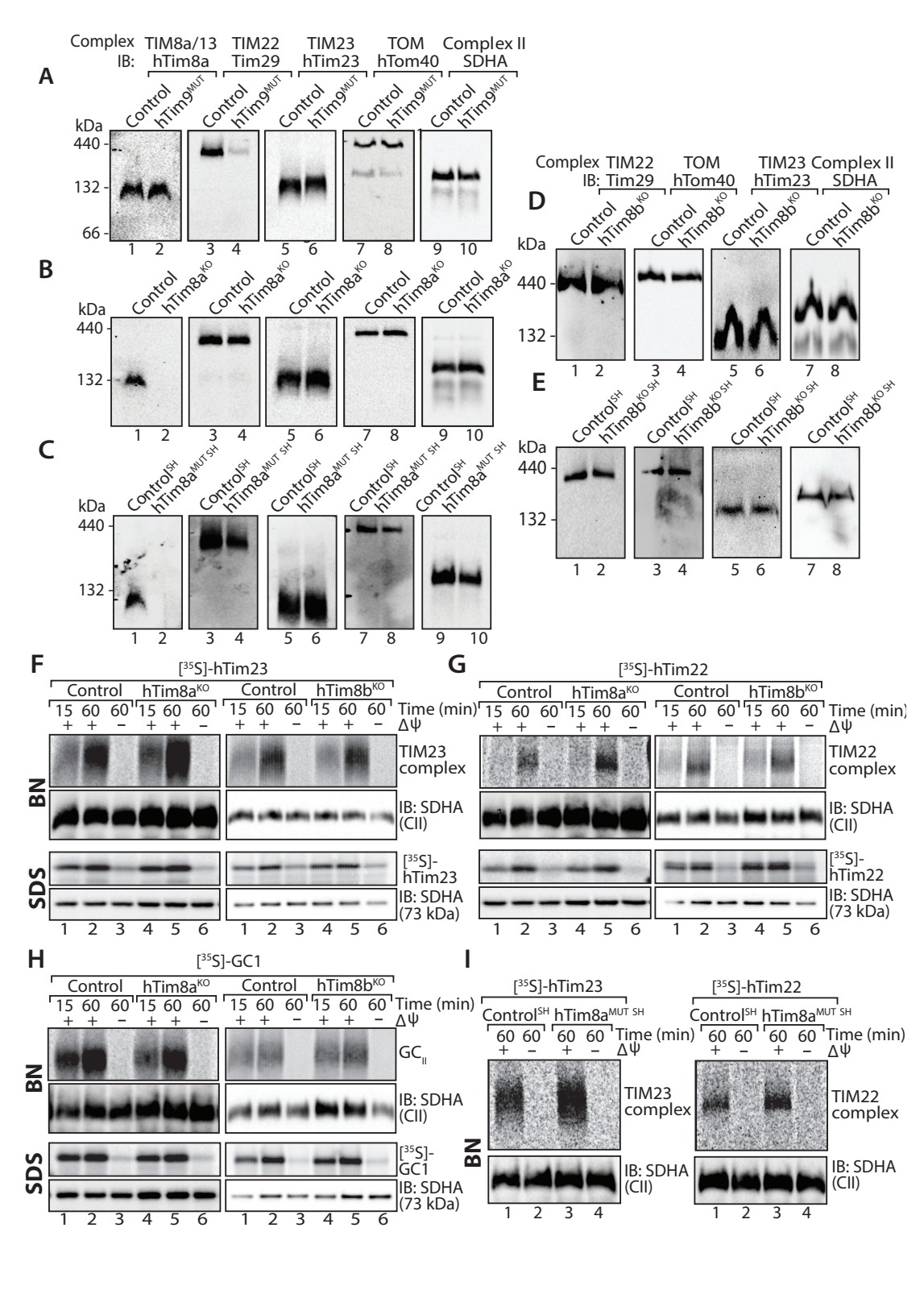

**Figure 1.** Cells lacking hTim8a have no defects in the TIM22, TIM23 or carrier biogenesis pathway. (A–E) Mitochondria were isolated from control, (A) hTim9$^{MUT}$ HEK293 cells, (B) hTim8a$^{KO}$ HEK293 cells, (C) hTim8a$^{MUT}$ SH-SY5Y cells, (D) hTim8b$^{KO}$ HEK293 cells, or (E) hTim8b$^{KO}$ SH-SY5Y cells prior to solubilisation in 1% digitonin-containing buffer. Mitochondrial lysates were subjected to Blue-Native electrophoresis prior to immunoblotting using the indicated antibodies. (F–H) [$^{35}$S]-hTim23, [$^{35}$S]-hTim22 or [$^{35}$S]-GC1 were incubated with mitochondria isolated from control and hTim8a$^{KO}$ or hTim8b$^{KO}$)

*Figure 1 continued on next page*

*Figure 1 continued*

HEK293 cells for the indicated time in the absence or presence of a mitochondrial membrane potential (ΔΨ) prior to Proteinase K treatment. Samples were separated by SDS-PAGE or solubilised in 1% digitonin-containing buffer and separated by BN-PAGE and visualised using autoradiography. (I) Mitochondria isolated from control SH-SY5Y and hTim8a$^{MUT\ SH}$ cells were incubated with [$^{35}$S]-hTim23 or [$^{35}$S]-hTim22 for 60 min in the presence or absence of membrane potential (ΔΨ) before Proteinase K treatment. Mitochondria were reisolated and solubilised in digitonin prior to BN-PAGE and subsequent immunoblotting.

The online version of this article includes the following figure supplement(s) for figure 1:

**Figure supplement 1.** Generation of CRISPR/Cas9 genome-edited cell lines.
**Figure supplement 2.** Loss of hTim8a or hTim8b has no defect on TIM or carrier biogenesis.

cytochrome *c* (*Figure 3A*, compare lanes 1 and 2 and quantification). These changes were not apparent in hTim8b$^{KO}$ mitochondria (*Figure 3A*, lanes 3 and 4 and quantification) and were not off-target effects, as complementation with hTim8a (hTim8a$^{KO+WT}$) restored the steady-state protein levels of Bax, cytochrome *c* and AIF (*Figure 3B*). Alterations in the abundance of cytochrome *c* and Bax were not due to large changes in their gene expression as measured by qPCR, while the impact on AIF was due to transcriptional regulation (*Figure 3—figure supplement 1A*). Treatment of control and hTim8a$^{KO}$ cells with staurosporine (STS), induced a faster release of cytochrome *c* from mitochondria in hTim8a$^{KO}$ cells (*Figure 3C*), suggesting these cells were more sensitive to apoptotic induction. Indeed, when we measured the rate of apoptotic induction in control, hTim8a$^{KO}$ and hTim8b$^{KO}$ cells we noted a higher frequency of death following STS treatment only in cells lacking hTim8a (*Figure 3D*; shown as % of cell death in *Figure 3—figure supplement 1A*). To test if loss of hTim8a specifically leads to mitochondrial-mediated apoptotic cell death, control, hTim8a$^{KO}$ and hTim8b$^{KO}$ cells were challenged with ABT-737, a BH3 mimetic inhibitor of Bcl-xL, Bcl-2 and Bcl-w and the rate of apoptosis was significantly increased in hTim8a$^{KO}$ cells and only slightly in cells lacking hTim8b (*Figure 3E*). We propose that HEK293 cells lacking hTim8a are in a primed for cell death state such that they can undergo a rapid death following cellular insult.

We next asked if SH-SY5Y cells lacking hTim8a were adopting the same primed for death state. Mitochondria isolated from hTim8a$^{MUT\ SH}$ cells had a similar increase in the level of cytochrome *c* relative to control cells, but only modest changes in the steady-state levels of AIF and Bax were observed (*Figure 3F*). hTim8a$^{MUT\ SH}$ cells were also more vulnerable to STS-induced apoptosis compared to control cells (*Figure 3G*), and treatment with ABT-737 also increased the rate of apoptosis in these cells. These effects were reversed when cells were pre-treated with a broad-spectrum caspase inhibitor QVD-OPh (*Figure 3H*). In agreement with this, we observed increased caspase-3 cleavage in hTim8a$^{MUT\ SH}$ cells when treated with STS or ABT-737. Caspase-3 processing was completely inhibited in the presence of QVD-OPh (*Figure 3I*, compare lanes 6 and 8 to 10 and 12), consistent with the inhibition of apoptosis. We also examined whether loss of hTim8a increased vulnerability to ferroptosis (*Dixon et al., 2012*; *Simon et al., 2000*; *Wu et al., 2018*), but treatment with specific ferroptosis inducers, including: (i) Erastin, an SLC7A11 inhibitor; (ii) (1S,3R)-RSL3, which inhibits glutathione peroxidase four and (iii) buthionine sulfoximine (BSO), which depletes glutathione (GSH), did not enhance cell death of hTim8a$^{MUT\ SH}$ cells (*Figure 3—figure supplement 1C*). Thus, we conclude that loss of hTim8a sensitises cells to Bcl-2-regulated and caspase-dependent, intrinsic cell death.

## Mitochondrial dysfunction and changes to mitochondrial metabolism in cells lacking functional hTim8a

We asked why cells lacking hTim8a and not hTim8b displayed vulnerability to intrinsic cell death and measured a number of parameters to ascertain cell and mitochondrial health in hTim8a$^{KO}$, hTim8b$^{KO}$, hTim8a$^{MUT\ SH}$ and hTim8b$^{KO\ SH}$ cells. Cell viability measured using trypan blue staining revealed HEK293 cells lacking hTim8b were less viable than HEK293 cells lacking hTim8a, while SH-SY5Y cells lacking hTim8a (hTim8a$^{MUT\ SH}$)were significantly less viable than control and hTim8b$^{KO\ SH}$ cells (*Figure 4A*). Incubation of cells with the membrane potential indicator, TMRM, indicated compromised membrane potential in HEK293 cells lacking hTim8a, but a more severe defect in cells lacking hTim8b (*Figure 4B*). Interestingly, loss of hTim8b in SH-SY5Y cells had no impact on the mitochondrial membrane potential, while those cells lacking functional hTim8a had a significant

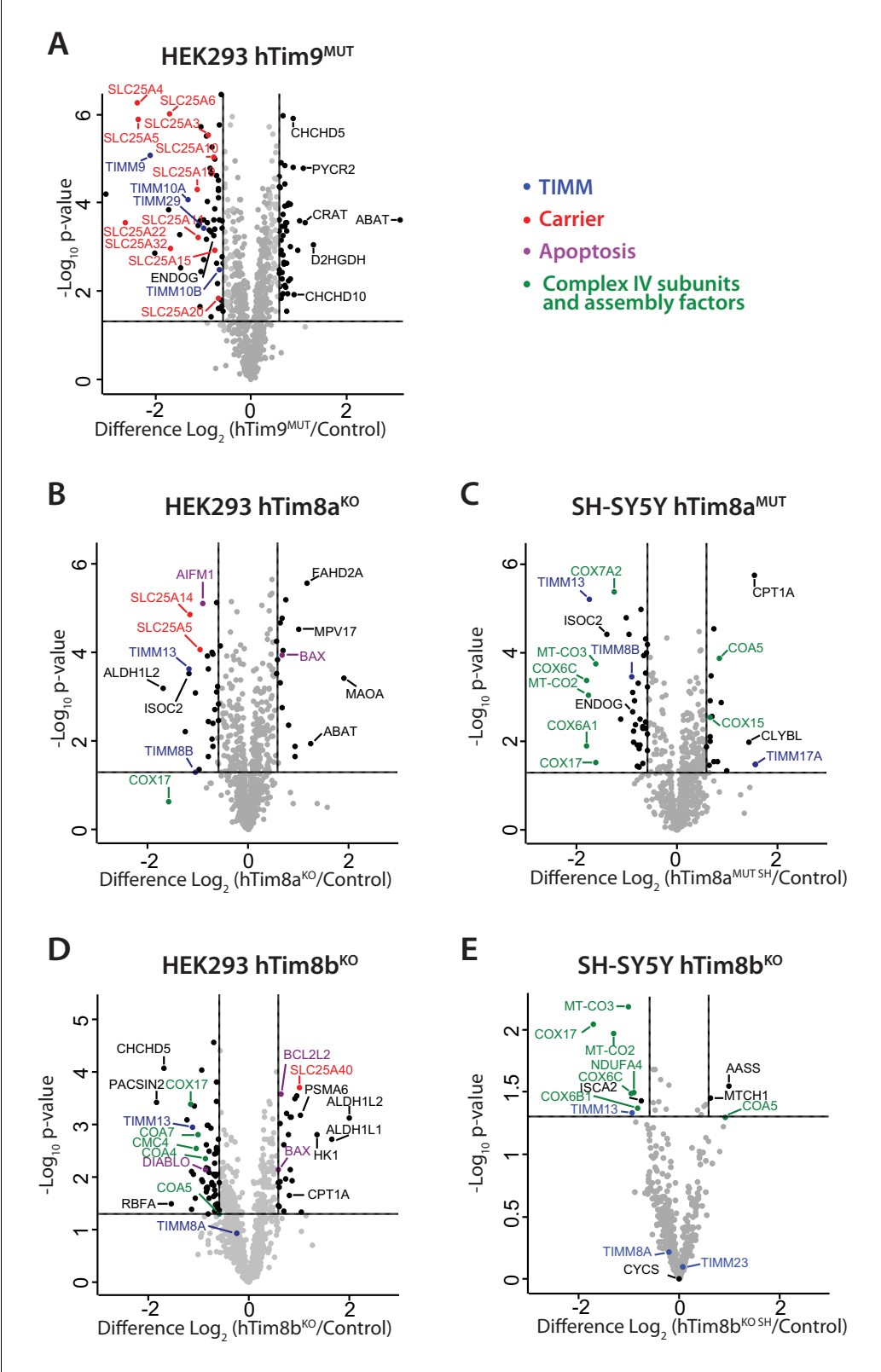

**Figure 2.** Loss of hTim8a shows cell-type specific consequences on the mitochondrial proteome. (A–E) Mitochondria were isolated from control and (A) hTim9[MUT] HEK293, (B) hTim8a[KO] HEK293, (C) hTim8a[MUT] SH-SY5Y, (D) hTim8b[KO] HEK293, or (E) hTim8b[KO] SH-SY5Y cells and subjected to label-free quantitative mass spectrometry analyses. Volcano plots showing relative levels of proteins in knock-out or mutant cells compared to control cells. n = 3

*Figure 2 continued on next page*

*Figure 2 continued*

biological replicates. Significantly altered proteins are located outside the line (p-value<0.05): TIMM proteins(blue), carrier proteins of the SLC25 family (red), apoptotic-related proteins (purple) and complex IV subunits and assembly factors (green) are indicated.

The online version of this article includes the following source data for figure 2:

**Source data 1.** Label-free quantitiative mass spectrometry on mitochondria isolated from: hTim9$^{MUT}$, hTim8a$^{KO}$, hTim8a$^{MUT\ SH}$,hTim8b$^{KO}$ and hTim8b$^{KO}$ $^{SH}$.

defect (*Figure 4B*). Oxygen consumption rate (OCR) measurements (*Figure 4C–4E*), suggested no respiration defect in HEK293 cells lacking hTim8a or hTim8b, however a minor reduction in maximal OCR in hTim8b$^{KO}$ HEK293 following the addition of FCCP was observed (*Figure 4C and D*, left panel). The extracellular acidification rate (ECAR) was significantly affected in the HEK293 lacking either hTim8a or hTim8b expression suggesting perturbations to glycolysis in these cells (*Figure 4D*, left panel). The OCR of hTim8a$^{MUT\ SH}$ was significantly compromised compared to control SH-SY5Y cells (*Figure 4C*, right panel). Both the basal and maximal respiration rates of hTim8a$^{MUT\ SH}$ were more severely impaired than hTim8b$^{KO\ SH}$(58% versus 78% of control; 60% versus 92% of control; *Figure 4D*, right panel). The activity of individual respiratory chain complexes was also measured along with citrate synthase (as a housekeeping enzyme). In agreement with the OCR measurements on HEK293 cells, we observed no defect in respiratory chain activity in hTim8a$^{KO}$ or hTim8b$^{KO}$ HEK293 cells (*Figure 4F*, left panel), instead respiratory complex activities seemed slightly upregulated in hTim8a$^{KO}$ HEK293, possibly to compensate for defective glycolysis observed in these cells. In contrast, the hTim8a$^{MUT\ SH}$ or hTim8b$^{KO\ SH}$ cells exhibited significant defects in the activities of Complex I and IV (55% and 67% of control; 64% and 81% of control), in agreement with the proteomics analyses showing clear defects on Complex IV biogenesis accompanied with mild defect in Complex I. This data suggests that hTim8b has a more prominent function in HEK293 cells, while the function of hTim8a is more important in the neuronal like SH-SY5Y cell line.

Targeted metabolite profiling of 200 polar metabolites in HEK293 and hTim8a$^{KO}$ HEK293 cells was undertaken using gas chromatography-mass spectrometry to further assess the consequences of hTim8a loss of function. Fifty-nine of these metabolites were significantly altered (BH-adjusted p<0.05) in hTim8a$^{KO}$ cells suggesting a distinct metabolomic footprint compared to control HEK293 cells (*Figure 5—source data 1*). Analysis of detected metabolites depicted as a heatmap with the top 40 most-significantly altered metabolites or as a pathway map (*Figure 5—figure supplement 1A*), suggested significant changes in central carbon metabolism in hTim8a$^{KO}$ cells, indicated by decreases in the levels of TCA cycle intermediates (including citrate, fumarate, malate), decreases in intermediates in lower glycolysis (DHAP, GAP, 2/3PGA, lactate) and a concomitant increase in intermediates in upper glycolysis and the pentose phosphate pathway (glucose, Glc6P, Rib5P, Ru5P). Loss of hTim8a therefore, leads to specific defects in mitochondrial metabolism and alterations of non-mitochondrial pathways of central carbon metabolism in HEK293 cells.

Metabolite profiling indicated that the metabolic response of SH-SY5Y cells to loss of hTim8a, differed from that of hTim8a$^{KO}$ HEK293 cells (*Figure 5A and B*; *Figure 5—figure supplement 2*). In particular, levels of many intermediates of the TCA cycle (citrate/isocitrate, fumarate, succinate) and interconnected pathways (glutamine) were significantly elevated, rather than reduced, in the hTim8a$^{MUT\ SH}$ cells (*Figure 5—figure supplement 2*). Both malate and aspartate were significantly reduced in hTim8a$^{MUT\ SH}$ cells (*Figure 5—figure supplement 2*), and the malate-aspartate shuttle is the most affected pathway in hTim8a$^{MUT\ SH}$ cells (*Figure 5B*), which is responsible for transferring NADH reducing equivalents from the cytosol to the mitochondria, and is very active in brain tissues (*McKenna et al., 2006*). hTim8a$^{MUT\ SH}$ cells also had a significant accumulation of epinephrine and changes in multiple metabolite intermediates of catecholamine pathway (metabolites/pathway coloured in pink in *Figure 5—figure supplement 2*). To investigate whether loss of hTim8a in SH-SY5Y cells was associated with a defect in mitochondrial metabolic fluxes and the malate-aspartate shuttle, control and hTim8a$^{MUT}$ SH-SY5Y cells were cultured in glucose-DMEM supplemented with [U-$^{13}$C]-glutamine, or glutamine-DMEM supplemented with [U-$^{13}$C]-glucose for 2 hr, and the time-dependent incorporation of $^{13}$C into the TCA cycle and related intermediates was quantified by GC-MS (*Figure 5C*, *Figure 5—source data 1*). The catabolism of $^{13}$C-glucose and $^{13}$C-glutamine was highly compartmentalized in both cell lines, with $^{13}$C-glucose being catabolized via glycolysis to lactate, and $^{13}$C-glutamine being primarily catabolized in the TCA cycle (*Figure 5C*). Levels of $^{13}$C-

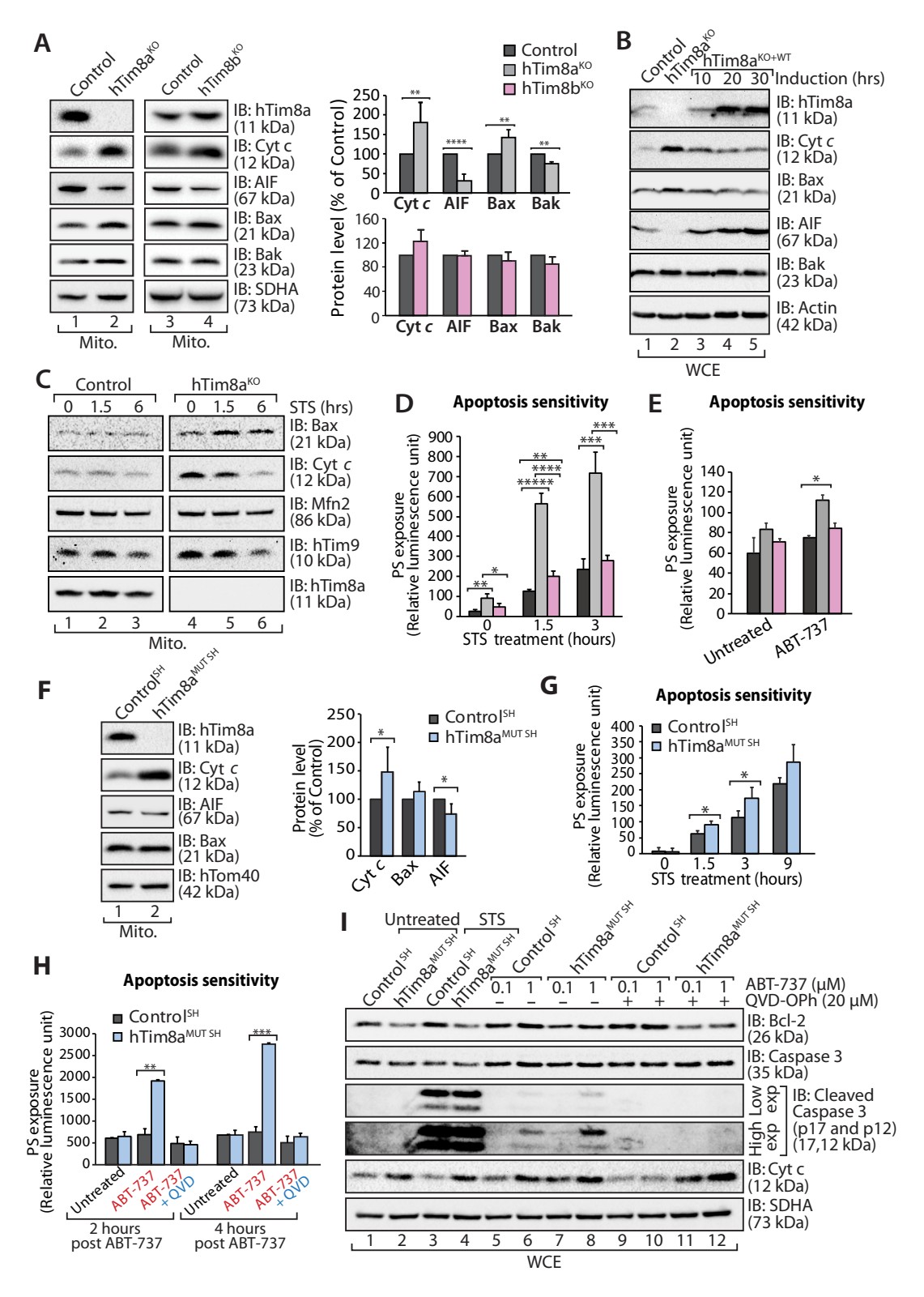

**Figure 3.** Lack of hTim8a, but not hTim8b, sensitises cells to apoptotic-cell death. (**A**) Mitochondrial lysates from control and hTim8a$^{KO}$ (left panel) or hTim8b$^{KO}$ (right panel) HEK293 cells were analysed by SDS-PAGE and western blotting. Relative protein levels of cytochrome $c$ (Cyt $c$), apoptosis-inducing factor (AIF), Bax and Bak were quantified and tabulated as mean ± SD (n = 3). **, p<0.01; ****, p<0.0001. (**B**) Cell lysates from control, hTim8a$^{KO}$ and hTim8a$^{KO}$ cells re-expressing hTim8a (hTim8a$^{KO+WT}$) were analysed using SDS-PAGE and immunoblotting with the indicated antibodies.
*Figure 3 continued on next page*

Figure 3 continued

(C) Control and hTim8a[KO] cells were treated with staurosporine (STS; 1.5 µM) for 0, 1.5 or 6 hours prior to mitochondrial isolation and analysis by SDS-PAGE. (D and E) Control, hTim8a[KO] and hTim8b[KO] HEK293 cells were (D) treated with staurosporine (STS; 1.5 µM) for 0, 1.5 or 3 hours, or (E) incubated with ABT-737 (0.1 µM) for 0 or 2 hours. The rate of apoptosis was calculated by measuring phosphatidylserine (PS) exposure to the outer leaflet of plasma membrane (relative luminescence unit). n = 4 (STS treatment); n = 3 (ABT-737 treatment); mean ± SD; *, $p<0.05$; **, $p<0.01$; ***, $p<0.001$; ****, $p<0.0001$; *****, $p<0.00001$. (F) Mitochondria isolated from control SH-SY5Y and hTim8a[MUT SH] cells were analysed by SDS-PAGE and immunoblotting. Graph shows the relative levels of hTim23, Bax, Cyt c and AIF quantified and represented as mean ± SD (n = 5). *, $p<0.05$. (G) Apoptotic sensitivity of control and hTim8a[MUT SH] cells was measured following staurosporine (STS; 1.5 µM) treatment by assessing phosphatidylserine (PS) exposure (relative luminescence unit). n = 4, mean ± SD; *, $p<0.05$. (H) Control and hTim8a[MUT SH] cells was either: (i) left untreated, (ii) treated with ABT-737 (0.1 µM) or (iii) pretreated with QVD-OPh (20 µM) for 20 min prior to ABT-737 treatment, for 2 or 4 hr prior to measuring cellular apoptotic sensitivity by assessing phosphatidylserine (PS) exposure (relative luminescence unit). n = 3, mean ± SD; **, $p<0.01$; ***, $p<0.001$. (I) Control and hTim8a[MUT] cells were either (i) left untreated, (ii) treated with STS (1.5 µM) or (iii) treated with ABT-737 (0.1 or 1 µM) with or without preincubation with QVD-OPh. Cell lysates were harvested following these treatments for SDS-PAGE and immunoblot analyses using the indicated antibodies.

The online version of this article includes the following figure supplement(s) for figure 3:

**Figure supplement 1.** Cells lacking fucntional hTim8a are not sensitised to ferroptosis-induced cell death, but intrinsic cell death.

enrichment in different intermediates in the two [13]C-glucose labelled cell lines were similar, with the exception of glycerol-3-phosphate, which was more highly labelled in hTim8a[MUT SH] (*Figure 5C*).

Glutamine feeds into the TCA cycle via the glutamine-glutamate-α-ketoglutarate pathway and can be used to drive mitochondrial respiration, and the synthesis of precursors for lipid and nucleotide biosynthesis, and neurotransmitters in neurons (*Plaitakis et al., 2017*). While levels of uptake of [13]C-glutamine were not affected in the Tim8a[MUT SH] cells (indicated by labelling levels of 5-oxoproline), levels of [13]C-enrichment in intermediates in the oxidative TCA cycle, starting from α-ketoglutarate (as reflected in glutamate labelling) to malate/oxaloacetic acid were decreased by ~30% compared to control cells (*Figure 5C*). Strikingly, [13]C-enrichment in citric acid was almost completely repressed in the [13]C-glutamine-fed hTim8a[MUT] cells, indicating that most or all of the oxaloacetate produced in the mitochondria is exported to the cytosol as part of the malate-aspartate shuttle. Loss of hTim8a in SH-SY5Y cells therefore appears to lead to a decrease in the rate of utilization of glutamine, the major carbon source driving the mitochondrial TCA cycle and oxidative phosphorylation, leading to decreased production of malate and oxaloacetate and the pool of intermediates for the malate-aspartate shuttle. These analyses indicate that loss of hTim8a in SH-SY5Y cells leads to dysregulation of mitochondrial metabolism and redox balance, indicating a vital role in neuronal cell biology.

## Dissecting the role of hTim8a and hTim8b in Complex IV biogenesis

The proteomics data in *Figure 2* suggested a role of hTim8a and hTim8b in Complex IV biogenesis. We wanted to dissect if the downstream consequences of hTim8a depletion, such as membrane potential defects and cell death sensitivity were in fact due to Complex IV biogenesis defects or some additional role of hTim8a. A closer look at the quantitative mass-spectrometry data from all four knock-out cell lines highlighted the importance of hTim8a and hTim8b in maintenance of Complex IV subunits, especially in the SH-SY5Y neuronal cell model (*Figure 6A and B*). hTim8b[KO] HEK293, hTim8a[MUT] SH-SY5Y and hTim8b[KO] SH-SY5Y cells showed decreased abundance in Complex IV subunits or assembly factors compared to control cells. hTim8a[KO] HEK293 did not share this phenotype, however this may reflect the tissue-specific expression and/or function of hTim8a. Further support for a novel role of hTim8 proteins in Complex IV assembly comes from the minimal effect that loss of either hTim8a or hTim8b had on SLC25A-family carrier proteins and TIM proteins, those thought to be putative substrates based on work in yeast (*Figure 6A*, upper panel). This contrasts with the broad defect in SLC25A proteins observed in hTim9[MUT] mitochondria (*Figure 2A*). By collating the relative abundance of subunits from each OXPHOS complex, it is clear the OXPHOS defect is not a generic OXPHOS defect or downregulation but specific to Complex IV (*Figure 6B*). Both hTim8a[MUT SH] and hTim8b[KO] SHSY5Y cell lines had reduced protein levels for all core subunits of Complex IV, a less severe reduction in the majority of Complex I subunits and little or no effect on Complexes II, III and V. Despite the downward trend of Complex I as a whole, few individual subunits are depleted significantly (*Figure 6A*, middle panel) and we believe this reflects a co-dependency of

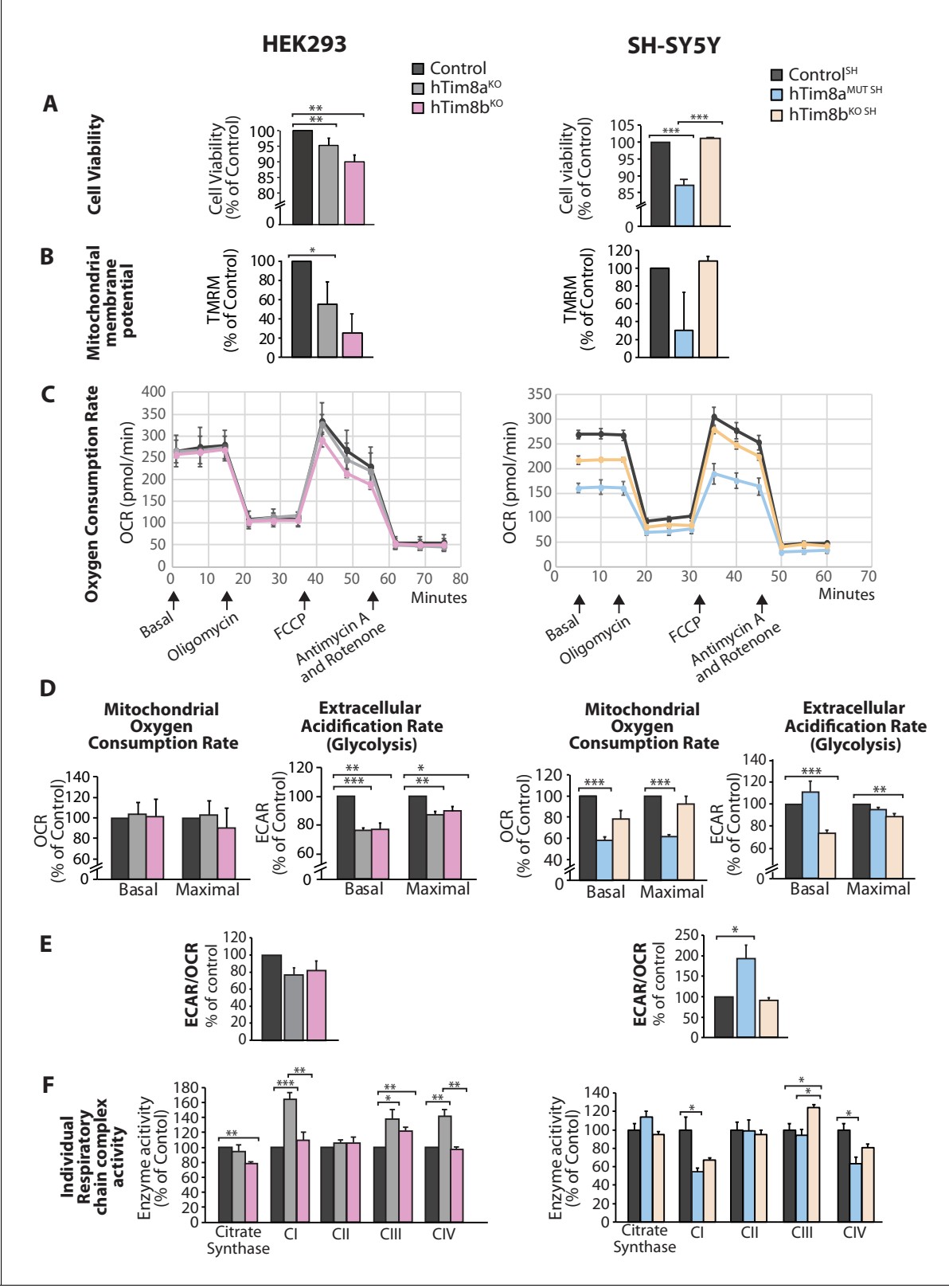

**Figure 4.** Cellular dysfunction in cells lacking hTim8a or hTim8b. (**A**) Cell viability of control and hTim8a[KO] or hTim8b[KO] HEK293 cells (left panel) or hTim8a[MUT SH] or hTim8b[KO SH] cells (right panel) was measured using trypan blue staining. Cells were seeded at the same confluency and 24 hours later were stained with trypan blue for cell counting. Viable cells were calculated as a percentage (1- [dead cells [stained blue]/total number of cells] X 100 = % viable cells). Data represents mean ± SD (n = 3). **, p<0.01, ***, p<0.001. (**B**) Mitochondrial membrane potential in control and hTim8a[KO] or

*Figure 4 continued on next page*

*Figure 4 continued*

hTim8b[KO] HEK293 cells (left panel) or hTim8a[MUT SH] or hTim8b[KO SH] cells (right panel) were quantified using TMRM uptake. Data represents mean ± SD (n = 3). *, p<0.05. (C) Mitochondrial oxygen consumption rate of control and hTim8a[KO]/hTim8b[KO] HEK293 cells (left panel) or hTim8a[MUT] or hTim8b[KO] SH-SY5Y cells (right panel) were measured using a Seahorse analyser. Oligomycin, FCCP, Antimycin A and Rotenone were added at the indicated time to the cells to allow for the measurement of basal, maximal and non-mitochondrial respiration rate. Error bars represent SEM (n=3). (D) The basal and maximal mitochondrial oxygen consumption rate (OCR; left panel) or extracellular acidification rate (ECAR; right panel) of hTim8a or hTim8b-edited cells relative to control were calculated and represented as mean ± SEM (n = 3). *, p<0.05, **, p<0.01, ***, p<0.001. (E) The ratio of ECAR/OCR of control and hTim8a or hTim8b-edited cells were calculated and were subsequently normalised to control and represented as mean ± SEM (n = 3). *, p<0.05. (F) The relative activity of individual respiratory chain complexes: Complex I-IV and citrate synthase in control and hTim8a[KO] or hTim8b[KO] HEK293 cells (left panel) or hTim8a[MUT] or hTim8b[KO] SH-SY5Y cells (right panel) were measured and represented as mean ± SEM (n = 3). *, p<0.05, **, p<0.01, ***, p<0.001.

Complex I on Complex IV for stability in respiratory supercomplexes (*Li et al., 2007*) not a direct influence of hTim8a or hTim8b on Complex I.

We confirmed these observations biochemically by analysing isolated mitochondria from control, hTim8a[MUT SH] and hTim8b[KO] cells via SDS-PAGE (*Figure 6—figure supplement 1A*) and also looking at Complex IV integrity by BN-PAGE (*Figure 6C and D*). The latter supported the proteomic data, with hTim8a[KO] HEK293 cells showing little impact on Complex IV stability (*Figure 6C*), while mitochondria isolated from control and hTim8a[MUT] SH-SY5Y cells displayed reduced levels of Complex IV (COX4 antibodies) and preferential interaction of COX4 with in a high MW intermediate (*Figure 6D*, upper panels) as has been seen in patients with mutations in genes encoding Complex IV assembly factors (*Lazarou et al., 2009*; *Stroud et al., 2015*). Probing for Complex I-containing supercomplex (with NDUFA9 antibodies) showed reduced levels of this complex in hTim8a[MUT SH] mitochondria. Interestingly, mitochondria from HEK293 cells lacking hTim8b (hTim8b[KO]) showed a similar profile when probed with COX4 (*Figure 6D*, left panel), that is reduced levels of monomeric Complex IV and preferential interaction of COX4 with a high MW intermediate. However, the absence of hTim8b in the SH-SY5Y cells had little impact on the Complex IV monomer. This data reinforces the cell specific functions of the hTim8 isoforms in Complex IV biology.

In vitro mitochondrial import assays of a number of Complex IV subunits and assembly factors did not reveal any obvious defects or intermediates (*Figure 6—figure supplement 1B and C*), however we noted that there was a common impact on steady state levels of COX17 in all cells lines lacking either hTim8a or hTim8b (*Figure 2* and *Figure 2—source data 1*), and this suggested a COX17-dependent process was being impacted in the absence of hTim8a and hTim8b. We confirmed the reduction in COX17 protein levels in SH-SY5Y cells depleted of hTim8a (*Figure 7A*) and investigated if hTim8a is required for the import and/or assembly of COX17 using in vitro import analysis. [$^{35}$S]-COX17 import into mitochondria isolated from hTim8a[MUT] and control SH-SY5Y cells showed normal kinetics by SDS-PAGE (*Figure 7B*), which was expected given that COX17 is an established Mia40 substrate (*Banci et al., 2010*). However, analysis of in vitro import reactions by BN-PAGE to monitor assembly of [$^{35}$S]-COX17 did suggest that COX17 assembly was perturbed in the absence of hTim8a (*Figure 7C*; ~40% less COX17 assembly compared to control). We set out to investigate if hTim8a was interacting with COX17 to accommodate a COX17-dependent process in mitochondria. hTim8a-[FLAG] was re-expressed in hTim8a[MUT SH] cells and mitochondria were isolated from these cells for downstream immunoprecipitation (IP) using FLAG antibodies. Under these conditions no interaction between hTim8a and COX17 was captured (*Figure 7D*). We questioned if the interaction was too transient to be captured using standard IP conditions. Furthermore, cell growth on galactose enhances oxidative capacity and increases Complex IV activity and expression (*Aguer et al., 2011*; *Rossignol et al., 2004*). Therefore, we cultured control SH-SY5Y and hTim8a[MUT SH] cells in galactose-containing media and modified our protocol to include cross-linking with dithiobis-succinimidyl propionate (DSP) prior to immunoprecipitation using FLAG antibodies (*Figure 7E and F*). Using this approach, we captured an interaction between hTim8a[FLAG] and COX17 by western blotting (*Figure 7F*). Control and hTim8a[MUT SH] cells cultured in glucose or galactose were then treated using the same crosslinking and IP approach, however elution fractions were processed for label free quantitative mass spectrometry. When grown in glucose-containing media the most enriched interacting partners for hTim8a were hTim8b and hTim13, in addition to Complex IV assembly factors (Coa4 and Coa7) and one Complex IV subunit (Cox6B1) (*Figure 7G* and *Figure 7—source data 1*).

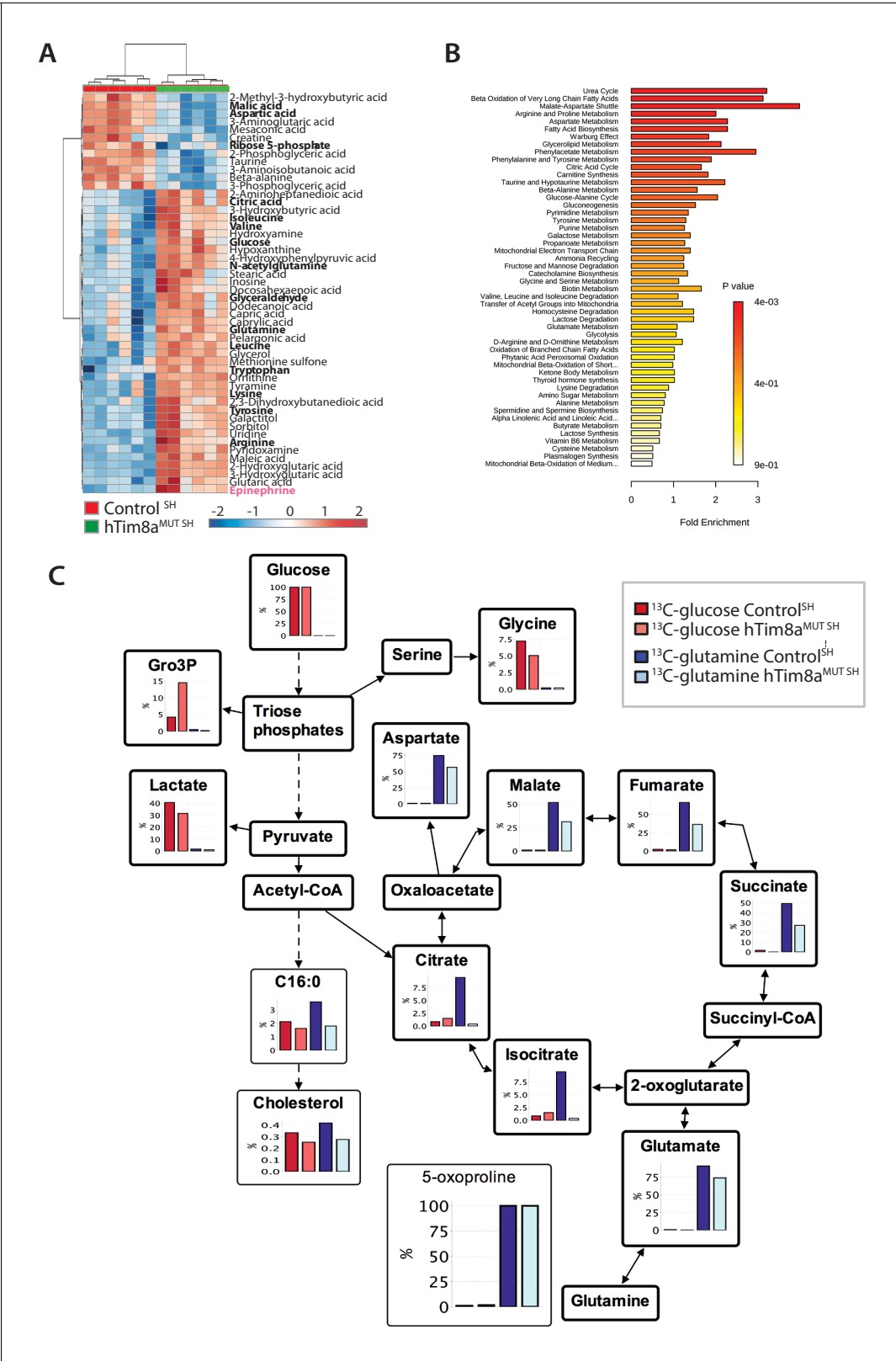

**Figure 5.** Metabolic rewiring in hTim8a^MUT SH-SY5Y cells. (**A**) Relative abundance of significantly affected metabolites in hTim8a^MUT compared to control SH-SY5Y cells. Heatmap representing the top 50 most affected metabolites extracted from control and hTim8a^MUT SH cells (p<0.05). Data represents values from six independent biological replicates of control and hTim8^MUT cells and fold changes are color-coded as indicated. See also *Figure 5—source data 1* and *Figure 5—figure supplement 2*. (**B**) Top 50 most affected cellular metabolic pathways in control and hTim8a^MUT SH cells

*Figure 5 continued on next page*

Figure 5 continued

were identified and represented as a clustered bar chart with fold enrichment indicated. The p-value of each of the metabolic pathways are color-coded as indicated. See also *Figure 5—source data 1* and *Figure 5—figure supplement 2*. (C) Percentage (%) of U-$^{13}$C-glucose (indicated by red bars) or U-$^{13}$C-glutamine (indicated by blue bars) labelled intracellular metabolites following a 2 hr incubation of control and hTim8a$^{MUT SH}$ cells in $^{13}$C-glucose or $^{13}$C-glutamine labelled media. Data represents the mean of 3 independent experiments and are mapped onto their respective metabolic pathway. See also *Figure 5—source data 2*.

The online version of this article includes the following source data and figure supplement(s) for figure 5:

**Source data 1.** Untargeted metabolomics profiling in Control, hTim8a$^{KO}$ and hTim8a $^{MUT SH}$ cells.

**Source data 2.** $^{13}$C-glucose and $^{13}$C-glutamine labelling of Control SH-SY5Y and hTim8$^{MUT SH}$ cells.

**Figure supplement 1.** Metabolomic profile in hTim8a$^{KO}$ HEK293 cells.

**Figure supplement 2.** Metabolomic profile in hTim8a$^{MUT}$ SH-SY5Y cells.

This profile changes under high respiratory demand and hTim13 is one of the least enriched partners, while a corresponding enrichment of Complex IV assembly factors (COA4, COA6, COA7) and subunits (COX6B1 and COX4l1), in addition to cytochrome *c* is observed (*Figure 7G*, right panel; *Figure 7—source data 1*). Additionally enriched proteins included key players of mitochondrial quality control pathways, pGAM5, OPA1, YME1L and the Prohibitins.

We asked if hTim8a is functioning in the copper transfer pathways required for Complex IV assembly and in which COX17 is a key player. Isolated mitochondria were pre-treated with either: (i) the copper chelator, bathocuproine disulfonate (BCS), or (ii) reduced glutathione (GSH) to reduce the disulfide bonds prior to crosslinking/immunoprecipitation (*Figure 7I*). The transient interaction between hTim8a and COX17 was not influenced by BCS suggesting that copper is not critical for their association. However, there was a pronounced reduction in the interaction of hTim8a$^{FLAG}$ and COX17 with COX4 upon copper chelation (*Figure 7I*, compare lanes 5 and 6), showing the validity of the approach. Importantly, pre-incubation of mitochondria with GSH significantly compromised the interaction of hTim8a$^{FLAG}$ with COX17 (*Figure 7I*, compare lanes 4 and 5), indicating that the oxidation status of the cysteine residues in these proteins is crucial to maintain their association. These data show that hTim8a interacts with a collection of Complex IV assembly factors and subunits and this interaction is enhanced under high respiratory demand, suggesting a novel role of hTim8a in the maturation of Complex IV, analogous to that of an assembly factor.

## Oxidative stress is linked to apoptotic vulnerability in cells lacking hTim8a

Complex IV dysfunction is linked to increased mitochondrial ROS and cellular toxicity (*Srinivasan and Avadhani, 2012*). To assess oxidative stress in cells lacking hTim8a, cells were either left untreated or pre-treated with the oxidative stress inducing agent, menadione prior to H$_2$O$_2$ ROS level measurement. ROS levels were significantly elevated compared to controls in both SH-SY5Y (*Figure 8A*) and HEK293 cells lacking hTim8a, but not hTim8b (*Figure 8—figure supplement 1A*), confirming enhanced oxidative stress in the absence of hTim8a and coinciding well with the activity of Complex IV in these cells (*Figure 4F*). We examined if the apoptotic vulnerability of cells lacking hTim8a was due to this observed oxidative stress. Challenging the cells with menadione alone increased the apoptotic vulnerability of hTim8a-deficient SH-SY5Y cells (*Figure 8B*), indicating that ROS accumulation and an impaired capacity to deal with ROS are contributing to the heightened apoptotic vulnerability in hTim8a$^{MUT SH}$ cells. Attenuating the oxidative stress in hTim8a$^{MUT}$ SH-SY5Y cells with the anti-oxidants Vitamin E (α-tocopherol; Vit E) that prevents lipid peroxidation in cellular membranes (*Traber and Atkinson, 2007*; *Traber and Stevens, 2011*), reduces the levels of cytochrome *c* (*Figure 8C*) and was accompanied by lowered sensitivity of these cells to apoptotic insult (*Figure 8D*). Although Vitamin E treatment did not restore the abundance of Complex IV subunits and assembly factors, MTCO1 and COX17 (*Figure 8—figure supplement 1B*), or monomeric Complex IV (*Figure 8—figure supplement 1C and D*), we did not observe any effect with Vitamin C, which functions as a water-soluble antioxidant (*Traber and Stevens, 2011*), suggesting a Vitamin E specific response.

Next we addressed if Complex IV disassembly and the resulting oxidative stress was the primary event causing the priming of hTim8a deficient cells for cell death. Given the common impact on COX17 across our cell models we depleted SH-SY5Y cells of COX17 via siRNA and monitored the

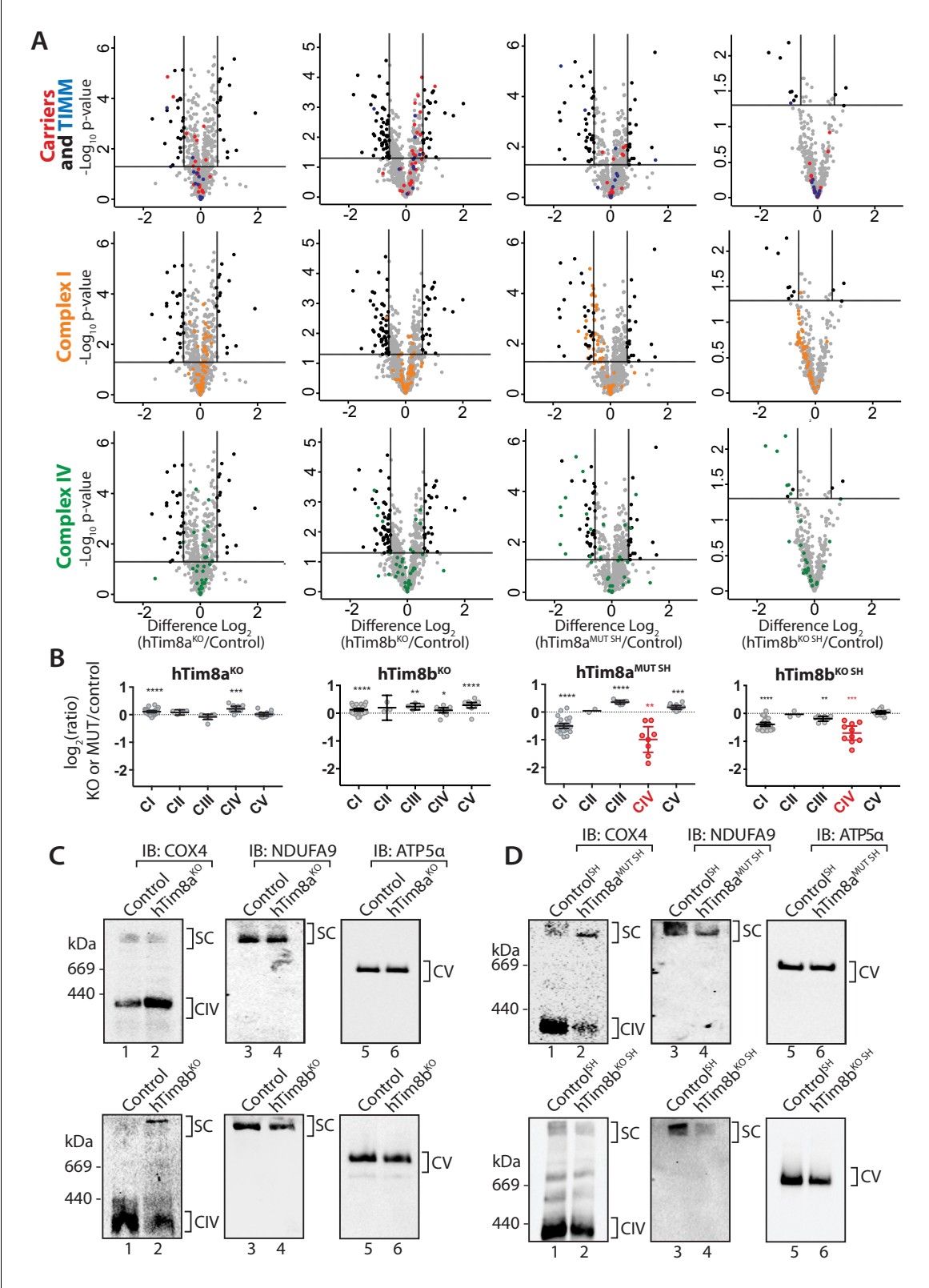

**Figure 6.** hTim8a and hTim8b have a cell type specific function in Complex IV biogenesis. (**A**) Data from label-free quantitative mass spectrometry shown in *Figure 2* was used to create volcano plots showing relative levels of proteins in CRISPR-edited cells compared to control cells. n = 3 biological replicates. Significantly altered proteins are located outside the line (p-value<0.05). TIM22 complex subunits/small TIM/hTim23 (blue), carrier proteins of the SLC25 family (red), Complex I subunits (orange) and Complex IV subunits and assembly factors (green) are highlighted and indicated in *Figure 6 continued on next page*

*Figure 6 continued*

separate plots. Refer to *Figure 2—source data 1*. (B) Relative abundance of respiratory chain complexes Complex I to V (subunits and assembly factors) in mitochondria isolated from control and hTim8a andhTim8b CRISPR-edited HEK293 or SH-SY5Y cells, detected using quantitative mass spectrometry in *Figure 2* were quantified and tabulated as mean ± SEM (n = 3). *, p<0.05, **, p<0.01, ***, p<0.001, ****, p<0.0001. (C and D) Mitochondrial lysates from control and hTim8a or hTim8b-CRISPR-edited HEK293 (C) or SH-SY5Y (D) were solubilised in 1% digitonin-containing buffer and analysed using BN-PAGE and immunoblotting with the indicated antibodies to assess for the stability of respiratory chain complexes. See also *Figure 6—figure supplement 1*.

The online version of this article includes the following figure supplement(s) for figure 6:

**Figure supplement 1.** Loss of hTim8a or hTim8b in SH-SY5Y cells leads to altered levels of multiple Complex IV subunits or assembly factors.

cellular consequences. Like SH-SY5Y cells lacking hTim8a, cells lacking COX17 had increased protein levels of cytochrome *c* (*Figure 9A*); destabilisation of Complex IV on BN-PAGE (*Figure 9B*); reduced mitochondrial membrane potential (*Figure 9C*); enhanced sensitivity to intrinsic cell death (*Figure 9D and E*); and oxidative stress (*Figure 9F*) that could initiate apoptosis (*Figure 9G*). Given this we reexpressed COX17$^{FLAG}$ in hTim8$^{MUT\ SH}$ cells to assess if the phenotypes observed in the mutant cells could be complemented by restoring the levels of COX17 (*Figure 9H*). Indeed, the parameters measured: (i) mitochondrial membrane potential (*Figure 9I*); (ii) apoptosis sensitivity following STS treatment (*Figure 9J*); (iii) oxidative stress (*Figure 9K*); and (iv) apoptosis sensitivity due to oxidative stress (*Figure 9L*) were all reverted in hTim8$^{MUT\ SH}$ cells expressing the COX17 (*Figure 9H*), suggesting that defective Complex IV and the corresponding oxidative stress is initiating downstream sensitisation of SH-SY5Y cells lacking hTim8a to death.

## Discussion

This is the first study to comprehensively analyse the function of TIM chaperones across different human cell models and our findings highlight the functional specialisation of mitochondrial chaperones in different tissues. Comparative analysis of neuronal-like and non-neuronal cells demonstrates the cell-specific function of the hTim8a and hTim8b isoforms, with hTim8a function being critical for SH-SY5Y cells. While loss of hTim9 resulted in profound disruption of the TIM22-biogenesis pathway, removal of hTim8a or hTim8b from both HEK293 or SH-SY5Y cells showed no impact to the biogenesis of the TIM22 complex or associated substrates, in particular hTim23, whose perturbed import has been suggested to be the underlying basis of MTS (*Rothbauer et al., 2001*). Rather, our data suggests that hTim8a and hTim8b have evolved novel functions in human mitochondria in particular in the biogenesis of Complex IV.

Complex IV dysfunction is associated with increased mitochondrial ROS and cellular toxicity and causes a clinically heterogeneous variety of neuromuscular and non-neuromuscular disorders in childhood and adulthood and can result from either nuclear or mitochondrial mutations (*Frazier et al., 2019*). Complex IV is unique among the respiratory chain complexes as it has tissue-specific subunit isoforms, which are believed to have regulatory roles in adaptation of tissues to specific metabolic demands (*Wallace and Fan, 2010*). Our crosslinking and IP data suggests that hTim8a and hTim8b likely have multiple functions in mitochondrial biology. Growth in glucose (low respiratory demand) shows a predominant interaction with: (i) partner proteins hTim8b and hTim13; (ii) Complex IV related proteins (Coa4, Coa7, cytochrome *c* and COX6B1) and key players involved in mitochondrial quality control including Opa1, the prohibitins and pGAM5. Indeed, these interactions are evident in the literature with hTim13 coming down as a member of the SPY complex, an inner membrane regulatory hub (*Wai et al., 2016*) and hTim8a and hTim13 immunoprecipitating with pGAM5 (*Holze et al., 2018*). Interestingly under high respiratory demand (growth on galactose) hTim8a shifts away from its interaction with hTim13 to exclusively engage in the Complex IV interaction network (including Coa4, Coa6, Coa7, COX41L and COX6B1) and mitochondrial quality control interactions. COX assembly is a modular process that includes a number of assembly factors (*Timón-Gómez et al., 2018*). Based on the lack of isolated defects in a specific Complex IV module in cells lacking hTim8a or hTim8b, and the predominant engagement of hTim8a with proposed assembly factors, including COX17 we can only postulate at this stage that the hTim8a/b are serving as assembly factors in Complex IV biogenesis, or have some regulatory role in chaperoning/guiding assembly factors to their place of action. Our identification of a copper-independent but redox-sensitive

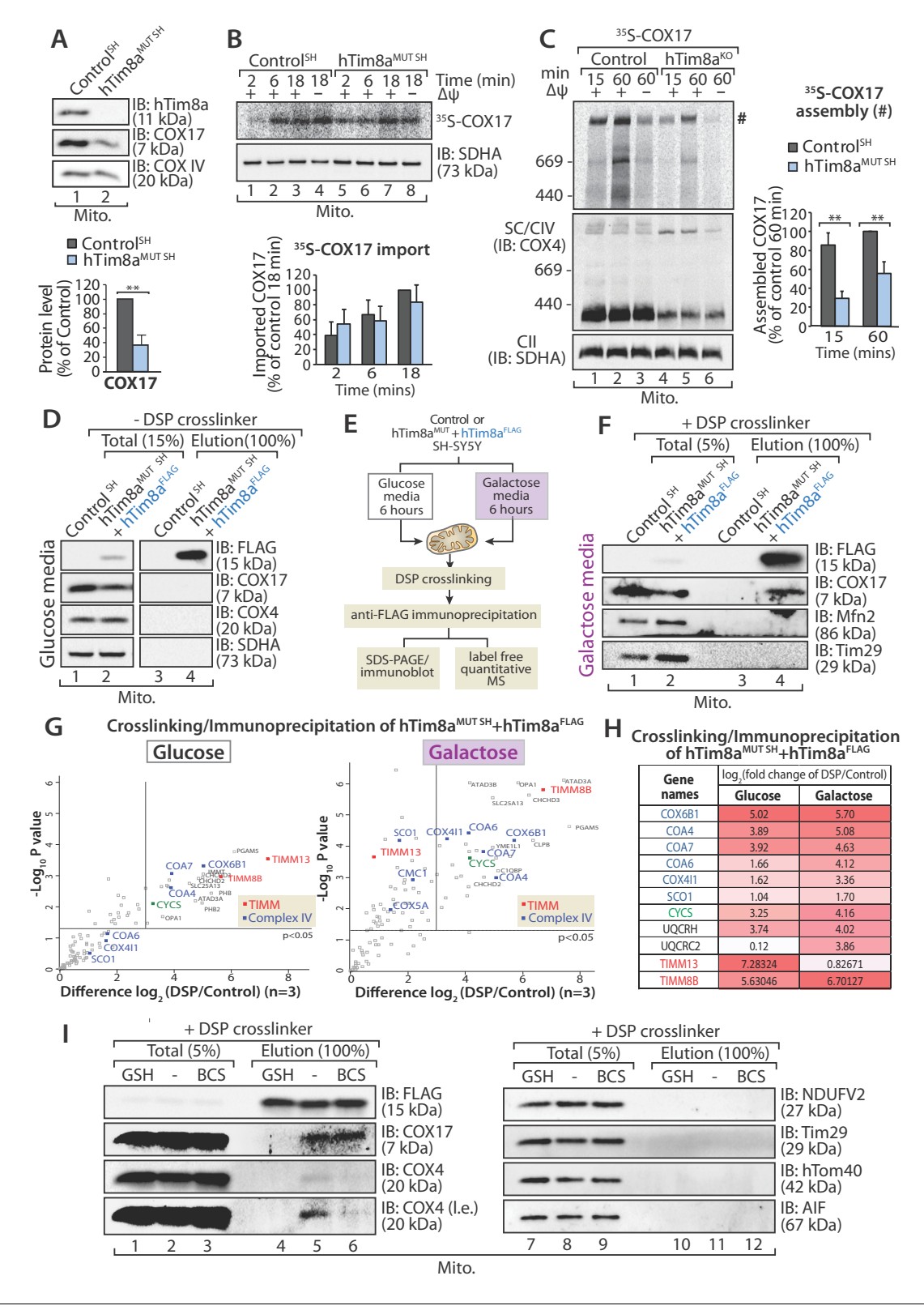

**Figure 7.** hTim8a functions in the early stage of assembly of Complex IV in SH-SY5Y cells. (**A**) Mitochondria isolated from control and hTim8a[MUT SH] cells were separated using SDS-PAGE and immunoblotted using antibodies against hTim8a, COX17 and COX4. Relative level of COX17 proteins in hTim8a[MUT] SH-SY5Y mitochondria compared to control were quantified and tabulated as mean ± SD (n = 3). **, p<0.01. (**B and C**) [35S]-labelled COX17 was incubated with mitochondria isolated from control and hTim8a[MUT SH] cells in the presence or absence of membrane potential (ΔΨ) and treated

*Figure 7 continued on next page*

*Figure 7 continued*

with proteinase K and PMSF prior to (**B**) TCA precipitation for SDS PAGE analysis, or (**C**) solubilised in 1% digitonin-containing buffer prior to BN-PAGE analysis, and autoradiography. The amount of (**B**) imported Cox17 (at 18 min) or (**C**) assembled COX17 (indicated by # on BN-PAGE at 60 min) were quantified and are represented as mean ± SD (n = 2 for (**B**) SDS-PAGE and n = 3 for (**C**) BN-PAGE). **, p<0.01. (**D**) Mitochondria were isolated from control or hTim8a$^{MUT}$ cells re-expressing hTim8a$^{FLAG}$ (hTim8a$^{MUT}$ + hTim8a$^{FLAG}$) SH-SY5Y cells and solubilised in 0.5% digitonin-containing buffer prior to immunoprecipitation and western blotting using the indicated antibodies. (**E**) Schematic representation of the assay performed to study the interaction between hTim8a and Complex IV. (**F**) hTim8a$^{FLAG}$ (hTim8a$^{MUT}$ + hTim8a$^{FLAG}$) SH-SY5Y were grown in galactose containing media. Mitochondria were isolated and treated for crosslinking with dithiobis-succinimidyl propionate (DSP) and immunoprecipitation with FLAG antibodies. Samples were separated by SDS-PAGE and probed with the indicated antibodies. (**G**) hTim8a$^{FLAG}$ and control cells were either grown in glucose (left) media or shifted to galactose media for 6 hr (right pabel) prior to mitochondrial isolation and crosslinking with dithiobis-succinimidyl propionate (DSP). Following immunoprecipitation samples were processed for label free quantitative mass spectrometry. n = 3 biological replicates. Significantly altered proteins are located outside the line (p-value<0.05). The small TIM chaperones are coloured in red; and Complex IV subunits or assembly factors are coloured in blue. (**H**) Table showing the relative fold of enrichment of hTim8a-crosslinked protein partners when cultured in glucose versus galactose media as illustrated in (**G**): Complex IV subunits/assembly factors (blue), Cytochrome *c* (CYCS, green), Complex III subunits/assembly factors (black) or hTim8a/13 (red). (**I**) hTim8a$^{MUT}$ + hTim8a$^{FLAG}$ mitochondria (protein expression induced with doxycycline for 20 hr followed by 6 hr incubation in galactose media) were either left untreated, subjected to bathocuproine disulfonate (BCS) chelation or reduced using glutathione (GSH) prior to DSP crosslinking and immunoprecipitation. Total and eluate fractions were separated using SDS-PAGE and analysed using immunoblotting.

The online version of this article includes the following source data for figure 7:

**Source data 1.** Interacting partners of hTim8b in SH-SY5Y cells in glucose versus galactose media.

interaction between hTim8a and COX17 hints that hTim8a could be a player that take parts in the redox-dependent modulation of COX17 for copper delivery to Complex IV. In line with our observations, loss of functional COA6, which is thought to be required for delivery of Cu(I) to COX2 (*Pacheu-Grau et al., 2015*; *Stroud et al., 2015*), results in a > 1.5 fold increase in the levels of COX17, hTim8a, hTim8b and hTim13 (*Stroud et al., 2015*). Intriguingly, a similar pattern of Complex IV subunit turnover is observed in cells lacking COA6 to that observed in hTim8a$^{KO}$ SH-SY5Y cells (*Stroud et al., 2015*), suggesting the proteins impact Complex IV at a similar stage of its assembly.

Cell lines lacking hTim8a display heightened vulnerability to apoptotic induction due to oxidative stress. This phenotype can be partially restored by the re-expression of COX17 in these cells, strongly suggesting that oxidative stress is the driver of the observed apoptotic vulnerability. We cannot dismiss that there may be additional players, potential substrates of hTim8a, that could be contributing to this phenotype. Indeed, while cells lacking hTim8b have reduced levels of COX17 these cells do not display the same level of oxidative stress and ultimately are not as sensitive to apoptotic induction. We propose that cells lacking hTim8a are primed for cell death, a term used to describe the proximity of cells to the apoptotic threshold (*Ni Chonghaile et al., 2011*). The apoptotic sensitivity of cells lacking hTim8a was selectively rescued by treatment with the lipid-soluble antioxidant Vitamin E, but not the water-soluble Vitamin C. Vitamin E plays a major role in protecting membranes and nervous tissues from oxidative stress, where it scavenges peroxyl radical to maintain the integrity of long chain polyunsaturated fatty acids in cellular membranes (*Traber and Atkinson, 2007*; *Traber and Stevens, 2011*). Our results therefore suggest lipid peroxidation as a potential oxidative pathway modulated upon hTim8a loss, leading to oxidative stress-induced apoptosis. Importantly, MTS is characterized in its end-stage by widespread and severe neuronal cell death in the central nervous system (*Hayes et al., 1998*; *Merchant et al., 2001*; *Tranebjaerg et al., 2001*). In agreement with this, cells lacking hTim8a have significant defects in glucose catabolism and mitochondrial metabolism, both of which are evolutionarily linked to the regulation of cell death (*King and Gottlieb, 2009*). In SH-SY5Y cells, loss of hTim8a was associated with defects in glutamine catabolism and the operation of the malate-aspartate shunt, which may underlie global changes in redox balance and ROS production, further contributing to the acute severity of loss of hTim8a on neuronal function (*Mergenthaler et al., 2013*). Changes to the malate-aspartate shunt coincides with work showing that citrin and aralar1, calcium-responsive aspartate-glutamate carriers that function in the malate-aspartate shuttle, are substrates of the hTim8a-hTim13 complex (*Roesch et al., 2004*). Strikingly, loss of hTim8a in SH-SY5Y also skewed the levels of epinephrine (elevated levels) and dopamine (lowered levels), two principal neurotransmitters, which are synthesised by the catecholamine biosynthesis pathway, a reaction unique and essential to the central nervous system and brain (*Kobayashi, 2001*).

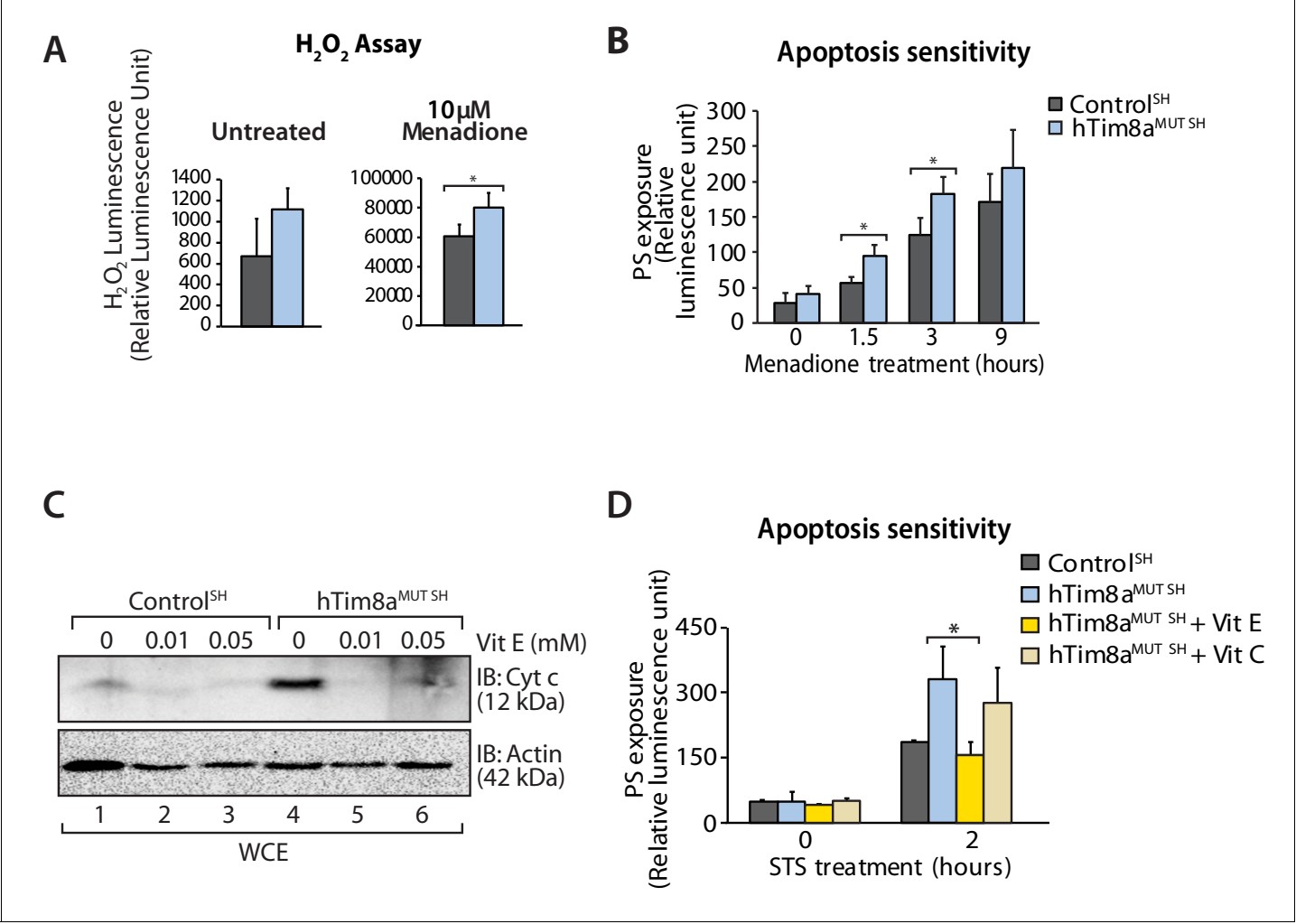

**Figure 8.** Elevated oxidative stress in hTim8a[MUT] SH-SY5Y cells sensitises cells to apoptosis. (**A**) Reactive $H_2O_2$ species present in untreated or menadione-pretreated (2 hr) control and hTim8a[MUT] SH-SY5Y cells were quantified using ROS-Glo $H_2O_2$ Assay. Data are shown as mean ± SD (n = 3, untreated; n = 4, menadione treatment). *, $p<0.05$. (**B**) Apoptotic sensitivity of control and hTim8a[MUT] SH-SY5Y cells was measured following menadione treatment for the indicated time, by assessing phosphatidylserine (PS) exposure (relative luminescence unit). Data is represented as mean ± SD. n = 4 for control and n = 3 for hTim8a[MUT SH]. *, $p<0.05$. (**C**) Mitochondria were isolated from control and hTim8a[MUT SH] cells following 24 hr of Vitamin E (Vit E, 0, 0.01 or 0.05 mM) treatment prior to SDS-PAGE and immunoblotting analyses. (**D**) Apoptotic sensitivity of control and hTim8a[MUT] cells was measured following Vit C (0.2 mM) or Vit E (0.01 mM) treatment by assessing phosphatidylserine (PS) exposure (relative luminescence unit). n = 3, mean ± SD; *, $p<0.05$.

The online version of this article includes the following figure supplement(s) for figure 8:

**Figure supplement 1.** Apoptotic sensitivity in the absence of hTim8a or hTim8b and the effects of Vitamin E treatment.

Our data demonstrates a cell-specific function for hTim8a in the assembly of Complex IV in the SH-SY5Y cell model, an in vitro model for neuronal cell function. Previous work by *Tranebjaerg et al. (2001)* highlighted neuronal loss as a prominent feature in MTS patients. We now provide insight into the molecular features eliciting this cell loss and show that cells lacking hTim8a are sensitised to an oxidative stress induced intrinsic cell death. Importantly, our data highlights a strategy for early therapeutic intervention using Vitamin E for mitochondrial neuropathologies like MTS. Given the role of hTim8a in Complex IV assembly we believe it would be appropriate to reclassify MTS from a secondary mitochondrial disease, to a primary mitochondrial disease.

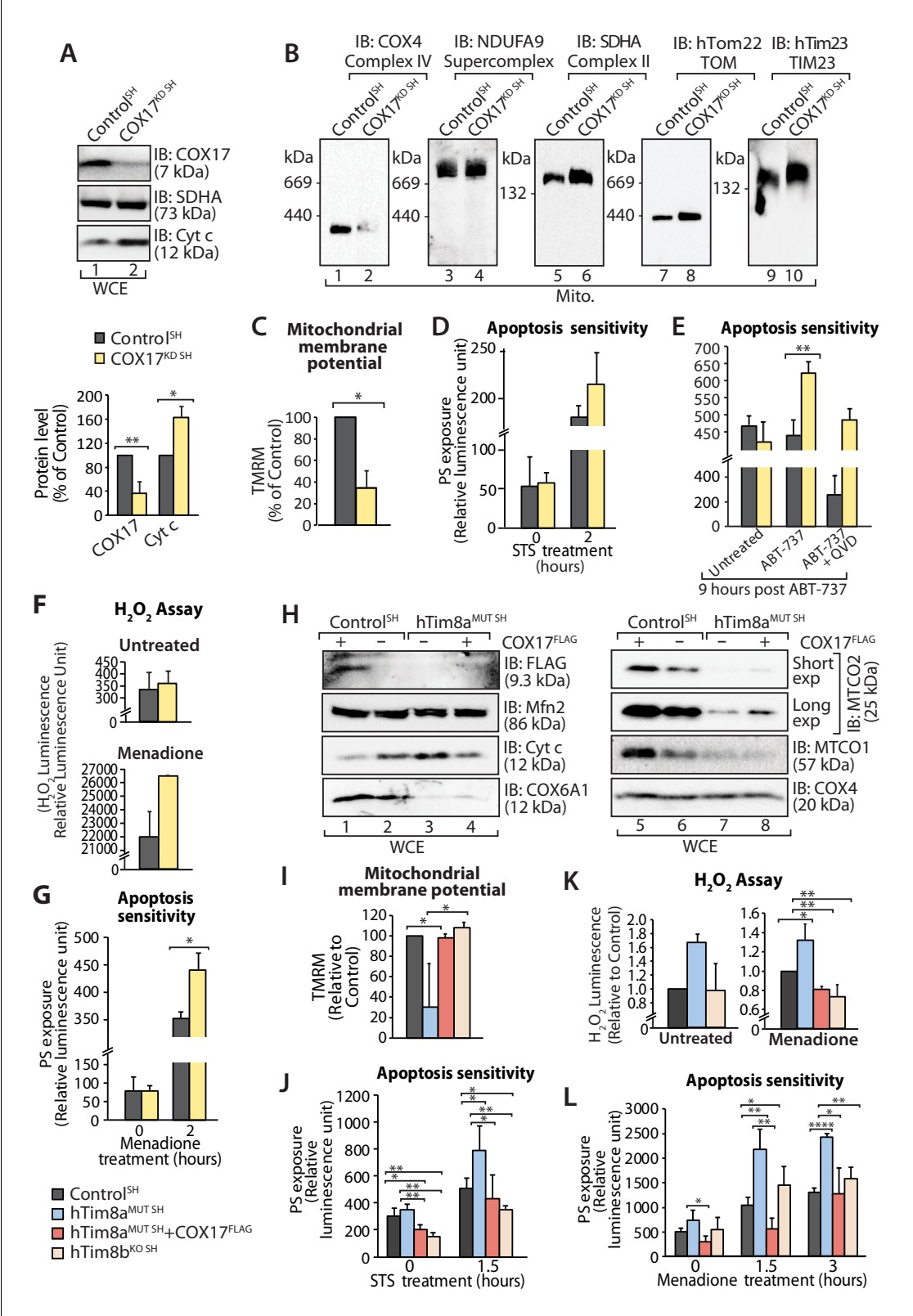

**Figure 9.** Loss of COX17 in SH-SY5Y cells leads to Complex IV defects and primes cells to apoptotic-cell death. (**A**) Cell lysate from SH-SY5Y cells depleted of COX17 (COX17[KD SH]) using siRNA were analysed using SDS-PAGE and immunoblotted with the indicated antibodies. Relative protein levels of COX17 and cytochrome *c* (Cyt *c*) were quantified and tabulated as mean ± SD (n = 3). *, p<0.05, **, p<0.01. (**B**) Mitochondria isolated from (**A**) were solubilised in 1% digitonin-containing buffer prior to BN-PAGE/western blot analyses using the indicated antibodies. (**C**) Control and COX17

*Figure 9 continued on next page*

*Figure 9 continued*

siRNA knockdown cells were stained using TMRM and mitochondrial membrane potential was measured and is represented as mean ± SD (n = 4). *, p<0.05. (D) Control and COX17$^{KD\ SH}$ cells were treated with staurosporine (STS; 1.5 µM) for 0 or 2 hr. Rate of apoptosis was calculated by measuring phosphatidylserine (PS) exposure to the outer leaflet of plasma membrane (relative luminescence unit). n = 3, mean ± SD. (E) Control or COX17$^{KD\ SH}$ cells were either: (i) left untreated, (ii) treated with ABT-737 (0.1 µM) or (iii) pretreated with QVD-OPh (20 µM) for 20 min prior to ABT-737 treatment, for 9 hr prior to measuring cellular apoptotic sensitivity by assessing phosphatidylserine (PS) exposure (relative luminescence unit). n = 3, mean ± SD. **, p<0.01. (F) Reactive H$_2$O$_2$ species present in untreated or menadione-pretreated (2 hr) control or COX17$^{KD}$ SH-SY5Y cells were quantified using ROS-Glo H$_2$O$_2$ Assay. Data are shown as mean ± SD (n = 4 for untreated, n = 3 for menadione-treatment). (G) Control or COX17$^{KD}$ SH-SY5Y cells were challenged with Menadione for the indicated time prior to measuring cellular apoptotic sensitivity by assessing phosphatidylserine (PS) exposure (relative luminescence unit). n = 3, mean ± SD. (H) Cell lysates from control and hTim8a$^{MUT}$ cells re-expressing COX17$^{FLAG}$, were analysed using SDS-PAGE and immunoblotted using the indicated antibodies. (I) Control, hTim8a$^{MUT}$, hTim8a$^{MUT}$ re-expressing COX17$^{FLAG}$ or hTim8b$^{KO}$ SH-SY5Y cells were assessed for mitochondrial membrane potential using TMRM uptake. n = 3, mean ± SD. *, p<0.05. (J) Control, hTim8a$^{MUT}$, hTim8a$^{MUT}$ re-expressing COX17$^{FLAG}$ or hTim8b$^{KO}$ SH-SY5Y cells were treated with staurosporine (STS, 1.5 µM) for 0 or 1.5 hr prior to measurement of the rate of phosphatidylserine (PS) exposure as an indication of the rate of apoptosis. Data were represented as mean ± SD, n = 4 for control and hTim8a$^{MUT\ SH}$ re-expressing; n = 3 for hTim8a$^{MUT\ SH}$ and hTim8b$^{KO\ SH}$ *, p<0.05; **, p<0.01. (K) Reactive H$_2$O$_2$ species present in untreated or menadione-pretreated (10 µM, 2 hr) control, hTim8a$^{MUT}$, control re-expressing COX17$^{FLAG}$, hTim8a$^{MUT}$ re-expressing COX17$^{FLAG}$ or hTim8b$^{KO}$ SH-SY5Y cells were quantified using ROS-Glo H$_2$O$_2$ Assay. Data are shown as mean ± SD (n = 3). *, p<0.05; **, p<0.01. (L) Apoptotic sensitivity of control, hTim8a$^{MUT}$, hTim8a$^{MUT}$ re-expressing COX17$^{FLAG}$ or hTim8b$^{KO}$ SH-SY5Y cells was measured following menadione (10 µM) treatment for 0, 1.5 or 3 hr, by assessing phosphatidylserine (PS) exposure (relative luminescence unit). n = 3 for control and hTim8a$^{MUT\ SH}$ re-expressing COX17$^{FLAG}$; n = 4 hTim8a$^{MUT\ SH}$ and hTim8b$^{KO\ SH}$, mean ± SD; *, p<0.05; **, p<0.01; ****, p<0.0001.

## Materials and methods

### Cell lines and culturing, siRNA transfection, transient protein expression and stable cell line generation

Cell lines used in this work were HEK293T, Flp-In T-REx 293 (Thermo Fisher Scientific) and SH-SY5Y. Cells were cultured at 37°C in Dulbecco's modified Eagle's medium (DMEM, Gibco) containing 5% or 10% [v/v] foetal bovine serum (FBS; In vitro Technologies) and 0.01% penicillin-streptomycin [v/v] under an atmosphere of 5% CO$_2$ and 95% air. For enhanced respiratory capacity cells were cultured in DMEM supplemented with 10 mM galactose, 10% dialyzed FBS, 1 mM sodium pyruvate, 50 µg/mL uridine, and 0.01% penicillin/streptomycin. Stable tetracycline inducible Flp-In T-REx HEK293 cell lines were generated using the T-REx system (Thermo Fisher Scientific) as previously described (*Kang et al., 2016*). Briefly, cells plated overnight at 37°C were transfected with pcDNA5/FRT/TO-[ORF] (plasmid encoding the ORF of interest) and pOG44 (encoding the Flp recombinase) at a 1:9 ratio (ng of DNA) using Lipofectamine 2000 prior to selection using 200 µg/ml of Hygromycin B (Thermo Fisher Scientific) for positive clones. To induce protein expression, cells were cultured in media supplemented with 1 µg/ml of tetracycline for the desired time. Stable SH-SY5Y cell lines were generated using retro-viral transduction system. Briefly, HEK293 cells were transfected with pBABE-[ORF]-puro and lentiviral packaging vectors (pVSV-G and pGag-pol) using Lipofectamine 3000. Viral supernatant was harvested at 48 hr post-transfection and layered onto SH-SY5Y cells in the presence of 8 µg/ml of polybrene. The transduced SH-SY5Y cells were selected using 0.5 µg/ml of puromycin for positive clones. For siRNA transfection, cells plated overnight were transfected with scrambled or *COX17* siRNA (5' GCAUGAGAGCCCUAGGAUU[dT][dT] 3'; Sigma). Briefly, 10 nM of siRNA was transfected into cells of choice using DharmaFECT (Dharmacon) according to the manufacturer's instructions. Cells were treated for a second transfection 48 hr post the first transfection and harvested at 72 hr post the initial transfection. Cell lines used in this study were free of mycoplasma contamination and were authenticated.

CRISPR/Cas9-gene editing and screening pSpCas9(*gRNA*)−2A-GFP containing guide RNA against (i) *TIMM8A*, (ii) *TIMM9* or (iii) *TIMM8B* were transfected into Flp-In T-REx 293 or SH-SY5Y for generation of CRISPR/Cas9-edited cells (*Ran et al., 2013*). Cells were subjected to single cell sorting for GFP positive expression prior to expansion and screening for CRISPR/Cas9 gene editing using: (i) western blotting with antibody against specific protein; (ii) allele sequencing to identify specific mutations introduced into the gene of interest and (iii) complementation of CRISPR/Cas9-edited cell line with wild type protein to ensure no off-target effects.

Genomic sequencing of CRISPR/Cas9-edited clones revealed:

| Gene-edited/cell line | Indel(s) introduced | Predicted protein impact | Frequency/depth |
|---|---|---|---|
| *TIMM8A/* Flp-In T-REx HEK293 (hTim8a^KO) | c.[55_56insGCAGGAGACGAAGGCCC TCATCTCTCGATGACGGGGGAATCTCG] | p.[L19Afs*21] | 16/16 |
| *TIMM9/* Flp-In T-REx HEK293 (hTim9^MUT) | c.[257del] c.[258-270del] | p.[L85Pfs*7] p.[Q87Pfs*7] | 5/10 5/10 |
| *TIMM8B/* Flp-In T-REx HEK293 (hTim8b^KO) | c.[2-82del] c.[4-6del; 77-112del] | p.0 p.0 | 5/7 2/7 |
| *TIMM8A/* SH-SY5Y (hTim8a^MUT SH) | c.[53_54InsT] Wildtype | p.[Q18Hfs*23] Wildtype | 6/10 4/10 |
| *TIMM8B/* SH-SY5Y (hTim8b^KO SH) | c.[2-18del] c.[4-6del] | p.0 p.0 | 5/9 4/9 |

## Mitochondrial isolation and cell lysate preparation

Mitochondria were purified from tissue culture cells using differential centrifugation as previously described (*Johnston et al., 2002*; *Kang et al., 2016*; *Kang et al., 2017*). Cell pellets were homogenised in isolation buffer (20 mM HEPES-KOH, pH 7.4, 220 mM mannitol, 70 mM sucrose, 1 mM EDTA, 0.5 mM PMSF, 2 mg/ml BSA) before centrifugation at 800 $g$ to isolate cellular and nuclear debris. The supernatant was centrifuged at 12,000 $g$ to obtain the mitochondria pellet. For preparation of whole cell lysate, cells were isolated and washed once in PBS prior to cell lysis in RIPA buffer (150 mM NaCl, 1% Triton-X100, 0.5% sodium deoxycholate, 0.1% SDS, 50 mM Tris-Cl, pH 8.0). Protein concentrations were quantified using Pierce BCA protein assay kit (Thermo Fisher Scientific) and desired amount of protein was TCA precipitated for SDS-PAGE analysis.

## Gel electrophoresis and immunoblot analysis

Tris-Tricine SDS-PAGE was carried out as described previously (*Kang et al., 2016*; *Kang et al., 2017*; *Schägger and von Jagow, 1987*). A 10–16% gradient was created using a gradient mixer. 16% and 10% acrylamide solutions (49.5% acrylamide: 1.5% bisacrylamide) were made up in tricine gel buffer (1 M Tris-Cl, pH 8.45, 0.1% [w/v] SDS) with 13% [v/v] glycerol added to the 16% mix. A stacking gel (4% acrylamide in tricine gel buffer) was overlayed onto the polymerised 10–16% gradient gel. Samples to be analysed were made up in SDS-PAGE loading dye (50 mM Tris-Cl, pH 6.8, 100 mM dithiothreitol, 2% [w/v] sodium dodecyl sulphate, 10% [w/v] glycerol, 0.1% [w/v] bromophenol blue). Electrophoresis was performed using Tris-tricine SDS-PAGE cathode buffer (0.1 M Tris, 0.1 M Tricine, pH 8.45, 0.1% SDS) and anode buffer (0.2 M Tris-Cl, pH 8.9).

Blue-Native-PAGE (BN-PAGE) was performed as previously described (*Ryan et al., 2001*; *Schägger, 1995*). 4–16% or 4–13% gradient gels were typically utilised. Acrylamide solution (49.5% acrylamide: 1.5% bisacrylamide) consisting of 13% [v/v] glycerol in BN gel buffer (66 mM α-amino n-caproic acid, 50 mM Bis-Tris, pH 7.0) was used to generate a gradient gel, which was polymerised using TEMED and APS. A 4% [v/v] acrylamide solution in BN gel buffer was overlaid onto the gradient gel. Samples were solubilised in solubilisation buffer (20 mM Tris-Cl, pH 7.4, 50 mM NaCl, 0.1 mM EDTA and 10% [v/v] glycerol) with 1% [w/v] digitonin and incubated on ice for 30 min. 1:10 vol of 10X BN loading dye (5% [w/v] Coomassie blue G250 (MP Biomedicals, LLC), 500 mM α-amino n-caproic acid, 100 mM Bis-Tris pH 7.0) was added to the clarified supernatant prior to loading. Electrophoresis was carried out using BN cathode buffer (50 mM tricine, 15 mM Bis-Tris, pH 7.0, 0.02% [w/v] Coomassie blue G 250) and anode buffer (50 mM Bis-Tris, pH 7.0).

Following electrophoresis, gels were transferred onto 0.45 μm PVDF membranes using a semidry transfer apparatus before being subjected to immunoblot analysis using specific primary antibodies and secondary antibodies (Sigma). Protein detection was performed using ECL chemiluminescent reagent (GE Healthcare) on the ChemiDoc MP imaging machine (BioRad). Quantification of western blot signal was performed following the manufacturer's instructions using the Image Lab software

(BioRad). Details of primary antibodies used for western blotting analyses are described in the resource table.

## Crosslinking and immunoprecipitation

For anti-FLAG immunoprecipitation, isolated mitochondria were solubilised in 0.5% digitonin-containing solubilisation buffer (0.5% digitonin [v/v], 20 mM Tris-Cl, pH 7.4, 150 mM NaCl, 1 X complete protease inhibitor (Roche)) for an hour, end over end at 4°C prior to a clarification spin at 16,000 g for 30 min. Supernatant was isolated and diluted in the same solubilisation buffer to a final of 0.1% digitonin. Samples was incubated to pre-equilibrated FLAG resin for an hour at 4°C (end-over-end). Resin with bound proteins was washed 4 X using solubilisation buffer containing 0.1% digitonin prior to elution using 0.2 M Glycine, pH 2.0.

For crosslinking/immunoprecipitation, mitochondria were resuspended in import buffer (20 mM HEPES-KOH (pH 7.4), 250 mM sucrose, 5 mM magnesium acetate and 80 mM potassium acetate) supplemented with 5 mM ATP at 1 mg/ml and incubated with the amino-group specific homobifunctional and cleavable cross-linker, dithiobis(succinimidylpropionate) (0.2 mM) (DSP; Thermo Fisher Scientific) end-over-end for 1 hr at 4°C. The reaction was quenched with 100 mM Tris-Cl, pH 7.4 for 30 min at 4°C. Mitochondria were re-isolated by centrifugation at 16,000 g for 20 min at 4°C and solubilised in lysis buffer (20 mM Tris-Cl, pH 7.4, 1 mM EDTA, 1% [w/v] SDS) with boiling at 95°C for 5 min. Samples were clarified by centrifugation at 16,000 g for 5 min at RT before being diluted in 1% TritonX-100-containing buffer (1% [v/v] TritonX-100, 20 mM Tris-Cl, pH 7.4, 150 mM NaCl, 1 X complete protease inhibitor (Roche)). The sample was added to pre-equilibrated FLAG-resin and incubated end-over-end for 1 hr at 4°C. The resin was isolated by centrifuging at 2,400 g for 3 min at 4°C and unbound proteins were isolated. The resin and bound proteins were washed X4 in 1% TritonX-100 buffer before resuspending into 2X loading dye for SDS-PAGE and immunoblotting analyses, or alternatively eluted with 0.2 M Glycine pH 2.0 for subsequent mass spectrometry.

## In vitro protein import and autoradiography

In vitro mRNA transcription and protein translation was performed using mMessage SP6 transcription kit (Ambion) and rabbit reticulocyte lysate system (Promega) respectively. In vitro translation was performed in the presence of $^{35}$S-methionine. Mitochondrial in vitro protein import assays were performed as described (*Ryan et al., 2001*). Radiolabeled proteins were incubated with isolated mitochondria resuspended at 1 mg/ml in import buffer (20 mM HEPES-KOH (pH 7.4), 250 mM sucrose, 5 mM magnesium acetate and 80 mM potassium acetate) supplemented with 10 mM sodium succinate, 1 mM 1,4-dithiothreitol and 5 mM ATP. Protein import was performed at 37°C for the desired amount of time in the presence or absence of membrane potential. To dissipate the membrane potential, a final concentration of 10 µM of Carbonyl cyanide-p-trifluoromethoxyphenyl-hydrazone (FCCP; Sigma-Aldrich) was added to each import reaction. Import reaction was stopped at 4°C followed by incubation with 50 µg/mL of PK and subsequent treatment with 1 mM PMSF. Mitochondria were re-isolated and TCA precipitated for SDS-PAGE analysis or resuspended into digitonin-containing buffer for BN-PAGE analysis. Autoradiography was performed to visualise the radioactive signal using a Typhoon Phosphor Imager (GE Healthcare). Radioactive images were processed using Image J software.

## Quantitative mass spectrometry and data analysis

For mass spectrometry (MS) analysis, isolated mitochondrial protein pellet (200 µg) was solubilised in sodium deoxycholate containing buffer (1% (w/v) SDC; 100 mM Tris-Cl, pH8.1; 40 mM chloroacetamide, 10 mM Tris(2-carboxyethy)phosphine (TCEP) followed by boiling at 99°C for 15 min and sonication (Powersonic 603 Ultrasonic Cleaner, 40 KHz on high power) for 10 min at RT. Overnight digestion with 1:100 [w/w] of trypsin was performed at 37°C. Samples were incubated with another 1:200 [w/w] trypsin for an additional 2 hr and tryptic peptides were extracted using 100 µL of ethyl acetate, 1% trifluoroacetic acid (TFA) and the entire volume was loaded onto 3MEmpore SDB-RPS stage tips (*Kulak et al., 2014*). The stage tips were washed twice using 200 µl of ethyl acetate, 1% TFA followed by two washes with 200 µL of 0.2% TFA. Peptides were eluted with 100 µL 80% acetonitrile, 5% ammonium hydroxide. All spins were performed at 1500 x g at RT. Eluates were dried using a SpeedVac concentrator and peptides reconstituted in 15 µL of 2% acetonitrile (ACN), 0.1%

TFA with sonication as above for 10 min. Samples were clarified at 16,000 *g* for 5 min at RT and soluble peptides transferred to autosampler vials.

Peptides were analysed by online nano-HPLC/electrospray ionization-MS/MS on Q Exactive Plus instruments connected to an Ultimate 3000 HPLC (Thermo-Fisher Scientific). For cell lines generated in the HEK293T background, peptides were loaded onto a trap column (Acclaim C18 PepMap nano Trap x 2 cm, 100 µm I.D, 5 µm particle size and 300 Å pore size; ThermoFisher Scientific) at 15 µL/min for 3 min before switching the pre-column in line with the analytical column (Acclaim RSLC C18 PepMap Acclaim RSLC nanocolumn 75 µm x 50 cm, PepMap100 C18, 3 µm particle size 100 Å pore size; Thermo Fisher Scientific). The separation of peptides was performed at 250 nL/min using a 128 min non-linear ACN gradient of buffer A [0.1% (v/v) formic acid, 2% (v/v) ACN] and buffer B [0.1% (v/v) formic acid, 80% (v/v) ACN]. Data were collected in positive mode using Data Dependent Acquisition using m/z 375–1575 as MS scan range, HCD for MS/MS of the 12 most intense ions with $z \geq 2$. Other instrument parameters were: MS1 scan at 70,000 resolution (at 200 m/z), MS maximum injection time 54 ms, AGC target 3E6, Normalized collision energy was at 27% energy, Isolation window of 1.8 m/z, MS/MS resolution 17,500, MS/MS AGC target of 2E5, MS/MS maximum injection time 100 ms, minimum intensity was set at 2E3 and dynamic exclusion was set to 15 s. For the cell line generated in the SH-SH5Y background, peptides were loaded onto a trap column (PepMap C18 trap column 75 µm x 2 cm, 3 µm, particle size, 100 Å pore size; Thermo Fisher Scientific) at 5 µL/min for 3 min before switching the pre-column in line with the analytical column (PepMap C18 analytical column 75 µm x 50 cm, 2 µm particle size, 100 Å pore size; Thermo Fisher Scientific). The separation of peptides for was performed at 300 nL/min using a 185 min non-linear ACN gradient of buffer A [0.05% (v/v) TFA acid, 2% (v/v) ACN] and buffer B [0.05% (v/v) TFA, 80% (v/v) ACN]. Data were collected in positive mode using Data Dependent Acquisition using m/z 375–1400 as MS scan range, HCD for MS/MS of the 15 most intense ions with $z \geq 2$. Other instrument parameters were: MS1 scan at 70,000 resolution (at 200 m/z), MS maximum injection time 50 ms, AGC target 3E6, Normalized collision energy was stepped at 25%, 30% and 35% energy, Isolation window of 1.6 m/z, MS/MS resolution 17,500, MS/MS AGC target of 5E4, MS/MS maximum injection time 50 ms, minimum intensity was set at 2E3 and dynamic exclusion was set to 30 s.

Raw files were analysed using the MaxQuant platform (*Tyanova et al., 2016a*) version 1.6.1.0 searching against the Uniprot human database containing reviewed, canonical and isoform variants in FASTA format (May 2016) and a database containing common contaminants. Default search parameters for a label-free (LFQ) experiment were used with modifications. Briefly, 'Match between runs' were enabled with default settings however. raw files from the two different instruments were given fraction numbers of 1 and 3 respectively to avoid spurious matching of MS1 peaks. Only unique peptides were used for quantification, using an LFQ minimum ratio count of 2. Using the Perseus platform (*Tyanova et al., 2016b*) version 1.6.1.1, proteins group LFQ intensities were Log2 transformed. Values listed as being 'Only identified by site', 'Reverse' or 'Contaminants' were removed from the dataset, as were identifications from <2 unique peptides. Annotations were imported from the IMPI Mitochondrial Proteome database (http://www.mrc-mbu.cam.ac.uk/impi) and 'known mitochondrial' proteins used for normalization of columns by row cluster. Experimental groups were assigned to each set of triplicates and a modified two-sided t-test based on permutation-based FDR statistics (*Tyanova et al., 2016b*) was performed. The negative logarithmic p-values were plotted against the differences between the $Log_2$ means for the two groups. A significance threshold (FDR < 0.05, s0 = 0.5) was used for all experiments. Profile plots were generated as per *Lake et al. (2017)*. Briefly, mean and standard deviations were calculated for each experimental group in Perseus. Values from a group with a standard deviation >0.6 were invalidated by conversion to 'NaN' and rows were filtered to contain at least two valid values in both experimental groups. Tables were imported into Prism seven software, following which a two-tailed ratio paired t-test was performed on the linearized Log2 LFQ Intensity mean values between the indicated groups.

## Metabolite extraction and GC-MS analysis

For steady-state metabolomics, cells were washed using phosphate-buffered saline (PBS) and frozen in liquid $N_2$ directly in the tissue culture plate. Metabolite extraction was performed using methanol: chloroform (9:1; [v/v]) in the presence of 0.5 nmol $^{13}C$-Sorbitol and 5 nmol $^{15}C_5$, $^{15}N$-valine as internal standards. Cells and supernatant were collected and centrifuged at 16,000 *g* at 4℃ for 5 min. Clarified supernatants were subjected to polar metabolite derivatization and analysis using a

Shimadzu GC/MS-TQ8040 system (*Best et al., 2018*). Comprehensive targeted metabolite profiling data was processed using Shimadzu GCMS Browser software to generate a data matrix and was statistically analysed (*Kang et al., 2017*). Differences between the test and control samples were calculated using Student's t-test to generate p-values that were further adjusted using Benjamini-Hochberg procedure to control for false-discovery rate (BH-adjusted p-value). To identify the relative flux of substrate into cells, $^{13}C_6$-glucose or $^{13}C_5$-glutamine labelling was performed as described (*Kang et al., 2017*; *Kowalski et al., 2015*). Briefly, cells were grown in glucose or glutamine-free media supplemented with 5 mM of $^{13}C_6$-glucose or 4 mM $^{13}C_5$-glutamine for 2 hr. Samples were then harvested for GC/MS and statistical analyses to determine the % of metabolite labelling as previously described. For metabolomic analyses, heatmaps and pathway mapping were generated using publicly available analysis tools: MetaboAnalyst 4.0 (*Chong et al., 2018*) or Vanted (*Rohn et al., 2012*).

## Gene expression with quantitative RT-PCR

Total RNA was extracted using NucleoSpin RNA kit (Macherey-Negal), cDNA was synthesised from 1 µg of RNA using High-capacity cDNA Reverse Transcription kit (Applied Biosystems) and gene expression was determined by SYBR-green RT-PCR on the Lightcycler 480 (Roche). Gene expression was normalised to GAPDH and analysed using the $\Delta\Delta C_t$ method. Primer sequences are detailed in the table below:

PCR primers for target genes:

| Gene | Forward (5'—3') | Reverse (5'—3') |
|------|-----------------|-----------------|
| BAX | CCATCATGGGCTGGACAT | CACTCCCGCCACAAAGAT |
| CYCS | CGGCGTGTCCTTGGACTTAG | CTTCCGCCCAAAGAGACCAT |
| AIFM1 | GGACTACGGCAAAGGTGTCA | CCTTGCTATTGGCATTCGGT |
| GAPDH | GGTGTGAACCATGAGAAG | CCACAGTTTCCCGGAG |

## Cellular viability measurements

Cell viability was assessed using trypan blue staining. $6 \times 10^5$ cells were seeded and allowed to grow at 37℃ for 24 hr. Cells were harvested and stained with 0.4% trypan blue solution (Thermo Fisher Scientific). Total number of cells and those stained blue were scored, and the number of viable cells was calculated by subtracting blue-stained (dead) cells from the total cell number. Conversely, % of viable cells was determined using the following formula: 1 - (Number of blue cells/ Number of total cells) x 100%. Statistical significance of triplicate experiments was determined using two-tailed Student's t-test.

The alamarBlue assay was carried out as previously described (*Liu et al., 2015*). This assay quantifies the conversion of non-fluorescent resazurin to fluorescent by mitochondrial metabolic activity. For these experiments, SH-SY5Y cells were seeded in 96-well plates ($1.5 \times 10^4$ cells/well) and allowed to adhere overnight. The next day, these cells were treated with ferroptosis inducers: BSO (Sigma-Aldrich), Erastin or (1S,3R)-RSL3 (SelleckChem). Following 72 hours of drug treatment (at the indicated concentrations), 20 µL of 20% (v/v) alamarBlue reagent (Thermo Fisher Scientific) was added to each well without removing the pre-existing media. Cells were incubated for 2 h at 37℃, and fluorescence was measured using a FLUOstar OPTIMA microplate reader (BMG Labtech) at an excitation of 540 nm and an emission of 590 nm. The percentage of viable cells was calculated as follows: $\frac{\bar{x}_D - \bar{x}_M}{\bar{x}_V - \bar{x}_M} \times 100$, where $\bar{x}_D$: means fluorescence of drug-treated wells, $\bar{x}_M$: means fluorescence of media only wells, and $\bar{x}_V$: means fluorescence of vehicle-treated control wells.

## Apoptosis assays

Apoptotic cell death was measured using Annexin V staining, which measures phosphatidylserine (PS) exposure on the outer plasma membrane as a measure of apoptosis. We utilised two different methods to measure the exposure of PS: (i) RealTime-Glo Annexin V Apoptosis and Necrosis Assay (Promega), which measures the relative rate of PS exposure (detected as relative luminescence unit)

and (ii) Fluorescence-activated cell sorting using flow cytometry which measure the % of cells that are positive for Annexin V staining.

For the RealTime-Glo Annexin V Apoptosis and Necrosis Assay, $4 \times 10^4$ cells were seeded onto a 96-well plate and incubated overnight at 37℃. The next day, cells were (i) incubated with fresh media supplemented with or without staurosporine (1.5 µM; Sigma-Aldrich); (ii) pretreated with or without QVD-OPh caspase inhibitor (20 µM; Sigma Aldrich) for 20 min before incubated with ABT737 (1 µM); (iii) incubated with fresh media supplemented with or without menadione (10 µM; Sigma-Aldrich); or (iv) pretreated with Vitamin C or E (0.2 mM; 0.01 mM; Sigma) for 24 hr before staurosporine incubation. The detection reagent was added to a final concentration of 1X according to the manufacturer's instructions. The relative fluorescence (necrosis) and luminescence (apoptosis) units were measured using FLUOstar OPTIMA microplate reader (BMG LABTECH) at the desired time interval following the different treatments. Differences in relative luminescence and fluorescence intensity between control and test samples were examined statistically using two-tailed Student's t-test.

To quantify the % of apoptotic cells using FACS assay, $3 \times 10^6$ cells were seeded the day before FACS analysis and incubated at 37 degrees 5% $CO_2$ overnight. Apoptosis was induced for 0, 3 or 6 hr with 1.5 µM Staurosporine. Following apoptosis induction, cells were harvested with trypsin and washed in Annexin V binding buffer (140 mM NaCl, 10 mM HEPES pH 7.4, 2.5 mM $CaCl_2$). The cells were centrifuged at 500 $g$ for 3 min and resuspended in the same binding buffer containing 1X Annexin V/ANXA5-FITC Apoptosis detection reagent (Abcam, ab14082) and 1 µg/mL Propidium Iodide (Merck, P4864). The cells were incubated at room temperature for 10 min in the dark. The cells were then pelleted at 500 $g$ for 3 min and were washed and resuspended in Annexin V binding buffer for FACS analysis. FACS analyses was done on a Becton Dickinson LSRII Flow Cytometer exciting with a 488 nm Laser and collecting the Annexin V FITC signal in a 530/30 nm detector and the Propidium Iodide in a 610/20 nm detector. Recorded 10,000–30,000 events per sample. Data analysis performed using FlowJo v10. A dot plot was generated using the area of (i) propidium iodide (to determine the relative proportion of viable and dead cells (%)) and (ii) Annexin V (to determine the relative proportion of apoptotic cells (%)).

## Reactive oxygen species measurement

Reactive oxygen species, $H_2O_2$ were measured using ROS-Glo $H_2O_2$ Assay according to the manufacturer's instructions (Promega). $4 \times 10^4$ cells were seeded onto a 96-well plate. Following 24 hr of growth, $H_2O_2$ dilution buffer and substrate (provided by the manufacturer; to generate luciferin precursor) with or without 10 µM menadione (Sigma-Aldrich) was added to the cells and was incubated at 37℃ for 2 hr incubation. ROS-Glo detection solution was added to the cells followed by a 20 min incubation at RT in the dark. Luminescence signals were measured using FLUOstar OPTIMA microplate reader (BMG LABTECH). Data generated from independent replicates of the experiment were analysed statistically using two-tailed Student's t-test.

## Mitochondrial membrane potential measurement

Mitochondrial membrane potential was measured using Tetramethylrhodamine, methyl ester (TMRM; ThermoFisher Scientific) staining using a microplate reader. Briefly, cells were seeded at $4 \times 10^4$ cells onto a 96-well plate and allowed to grow at 37℃ for 24 hr. Cells were treated with 75 nM TMRM stain and the fluorescence signal detected at 590 nm with an excitation at 544 nm using FLUOstar OPTIMA microplate reader (BMG LABTECH).

## OXPHOS enzymology

Enriched mitochondrial fractions were prepared from HEK293 and SH-SY5Y cell lines grown in triplicate under standard culture conditions and the activities of OXPHOS complexes were measured in samples with (CI, CII, CIV and CS) and without (CIII) hypotonic treatment, as described (*Frazier and Thorburn, 2012*).

## Mitochondrial oxygen consumption and extracellular acidification rate measurement using seahorse analyser

A Seahorse Bioscience XF24-3 Analyzer was used to measure oxygen consumption rates (OCR) and extracellular acidification rates (ECAR) in live cells according to manufacturer's procedures. Briefly, 50,000 (HEK293) or 100,000 (SH-SY5Y) cells were plated per well in XF24-3 culture plates treated with poly-D-Lysine (HEK293) or Matrigel/collagen (SH-SY5Y) and grown overnight under standard culture conditions. Rates were measured in non-buffered DMEM media. For HEK293 cells, each measurement cycle consisted of 2 min mix, 1 min wait and 3 min measure using the following inhibitors: 0.5 μM oligomycin; 0.1 μM carbonyl cyanide 4-(trifluoromethoxy) phenylhydrazone (FCCP); 0.5 μM rotenone; and 0.3 μM antimycin A. For SH-SY5Y cells, each measurement cycle consisted of 1 min mix, 1 min wait and 3 min measure using the following inhibitors: 0.5 μM oligomycin; 0.6 μM FCCP; 0.5 μM rotenone; and 0.3 μM antimycin A.

For each cell line, 3–7 replicate wells were measured in multiple plates (n = 3) and CyQuant (Life Technologies) was used to normalize measurements to cell number. Basal OCR and non-mitochondrial respiration (following rotenone and antimycin A addition) were calculated as an average of the three measurement points. Basal ECAR was calculated from the initial basal measurement cycle. To calculate maximal respiration, the initial measurement following FCCP addition was used, while maximal ECAR following oligomycin addition was calculated from the initial measurement (HEK293) or final measurement (SH-SY5Y).

## Antibodies

| Antibody | Company | Catalogue number |
|---|---|---|
| Rabbit polyclonal AGK | Atlas Antibodies | HPA020959 |
| Mouse monoclonal hTim9 | Abcam | ab57089 |
| Mouse monoclonal SDHA | Abcam | ab14715 |
| Rabbit polyclonal ANT3 | Abcam | ab154007 |
| Rabbit polyclonal Glutamate carrier 1 (GC1) | Abcam | ab137614 |
| Mouse monoclonal OXPHOS | Abcam | ab110413 |
| Mouse monoclonal ATP5alpha | Abcam | ab14748 |
| Rabbit polyclonal hTom70 | Abcam | ab83841 |
| Mouse monoclonal Cytochrome c | BD Biosciences | 556433 |
| Mouse monoclonal hTim23 | BD Biosciences | 611223 |
| Rabbit polyclonal COX4 | Cell Signaling Technology | 4850 |
| Mouse monoclonal MTCO1 (COX1) | Abcam | ab14705 |
| Mouse monoclonal MTCO2 (COX2) | Abcam | ab110258 |
| Rabbit polyclonal COX6A1 | Proteintech | 11460–1-AP |
| Rabbit monoclonal AIF | Cell Signaling Technology | 5318 |
| Rabbit polyclonal Caspase-3 | Cell Signaling Technology | 9662 |
| Rabbit polyclonal hTim8a | Proteintech | 11179–1-AP |
| Rabbit polyclonal hTim13 | Proteintech | 11973–1-AP |
| Rabbit polyclonal hTim44 | Proteintech | 13859–1-AP |
| Rabbit polyclonal COX17 | Sigma Aldrich | HPA048158 |
| Mouse monoclonal FLAG | Sigma-Aldrich | F1804 |

*Continued on next page*

*Continued*

| Antibody | Company | Catalogue number |
|---|---|---|
| Rabbit polyclonal hTim22 | Sigma-Aldrich | T8954 |
| Rabbit polyclonal Tim29 | Sigma-Aldrich | HPA041858 |
| Mouse monoclonal actin | Sigma-Aldrich | A2228 |
| Mouse monoclonal Bcl-2 | ThermoFisher Scientific | 13–8800 (Bcl-2–100) |
| Mouse monoclonal hTom22 | SantaCruz | Sc-58308 |
| Rabbit polyclonal NDUFV2 | Proteintech | 15301-1-AP |
| Rabbit polyclonal Sam50 | N/A | Mike Ryan, Monash University |
| Rabbit polyclonal NDUFAF2 | N/A | Mike Ryan, Monash University |
| Rabbit polyclonal hTom40 | N/A | Mike Ryan, Monash University |
| Rabbit polyclonal Mfn2 | N/A | Mike Ryan, Monash University |
| Rabbit polyclonal NDUFV2 | N/A | Mike Ryan, Monash University |
| Rabbit polyclonal NDUFA9 | N/A | Mike Ryan, Monash University |
| Rat monoclonal BAX | N/A | Grant Dewson, Walter and Eliza Hall Institute |

## Acknowledgements

YK is supported by Melbourne International Fee Remission Scholarship (MIFRS) and Melbourne International Research Scholarship (MIRS). DS is supported by a Research Fellowship from the Mito Foundation. We acknowledge funding from the Australian Research Council (DP170101249 to DS), NHMRC Project Grants (1125390, 1107094 to MTR., DRT., and DAS; 1140906 to DAS and MTR) and Fellowships (1140851 to DAS; 1022896 to DRT; and 1059530 to MJM), Victorian Government Department of Health and Human Services acting through the Victorian Cancer Agency (MCRF16002 to NJC), the Mito Foundation and the Victorian Government's Operational Infrastructure Support Program. We thank Simone Tregoning for research support. We thank the Bio21 Mass Spectrometry and Proteomics Facility and the Monash University Biomedical Proteomics Facility for the provision of instrumentation, training, and technical support. We acknowledge the use of the Biological Optical Microscopy Platform at the University of Melbourne. We thank A/Prof. Grant Dewson for discussion and reagents.

## Additional information

### Funding

| Funder | Grant reference number | Author |
|---|---|---|
| Australian Research Council | DP170101249 | Diana Stojanovski |
| Melbourne International Fee Remission Scholarship | | Yilin Kang |
| Melbourne International Research Scholarship | | Yilin Kang |
| Mito Foundation | Research Fellowship | Diana Stojanovski |
| NHMRC | | David R Thorburn Michael T Ryan |
| NHMRC | Project Grant 1125390 | David R Thorburn Michael T Ryan David A Stroud |
| NHMRC | Project Grant 1107094 | David R Thorburn Michael T Ryan David A Stroud |

| | | |
|---|---|---|
| NHMRC | Project Grant 1140906 | Michael T Ryan<br>David A Stroud |
| NHMRC | Fellowship 1140851 | David A Stroud |
| NHMRC | Fellowship 1022896 | David R Thorburn |
| NHMRC | Fellowship 1059530 | Malcolm J McConville |
| Victorian Cancer Agency | MCRF16002 | Nicholas J Clemons |

The funders had no role in study design, data collection and interpretation, or the decision to submit the work for publication.

### Author contributions
Yilin Kang, Data curation, Formal analysis, Investigation, Methodology, Writing—review and editing; Alexander J Anderson, Data curation, Investigation, Writing—review and editing; Thomas Daniel Jackson, Data curation, Formal analysis, Writing—review and editing; Catherine S Palmer, Data curation, Supervision, Investigation, Methodology, Writing—review and editing; David P De Souza, Data curation, Investigation, Methodology, Writing—review and editing; Kenji M Fujihara, Formal analysis, Investigation, Writing—review and editing; Tegan Stait, Investigation, Writing—review and editing; Ann E Frazier, Formal analysis, Supervision, Investigation, Writing—review and editing; Nicholas J Clemons, David R Thorburn, Michael T Ryan, Resources, Supervision, Writing—review and editing; Deidreia Tull, Data curation, Formal analysis, Supervision, Writing—review and editing; Malcolm J McConville, Resources, Supervision, Methodology, Writing—review and editing; David A Stroud, Resources, Data curation, Supervision, Investigation, Methodology, Writing—review and editing; Diana Stojanovski, Conceptualization, Resources, Formal analysis, Supervision, Funding acquisition, Writing—original draft, Project administration, Writing—review and editing

### Author ORCIDs
Catherine S Palmer (iD) http://orcid.org/0000-0002-5124-2400
Nicholas J Clemons (iD) http://orcid.org/0000-0001-9283-9978
Diana Stojanovski (iD) https://orcid.org/0000-0002-0199-3222

### Decision letter and Author response
Decision letter https://doi.org/10.7554/eLife.48828.sa1
Author response https://doi.org/10.7554/eLife.48828.sa2

# Additional files

### Supplementary files
• Transparent reporting form

### Data availability
All data generated or analysed during this study are included in the manuscript and supporting files. Source data files have been provided for Figure 2, Figure 5 and Figure 7.

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
