## [Decision Letter]

**Acceptance summary:**

The Mohr-Tranebjærg syndrome is a neurodegenerative disorder characterized by progressive hearing loss, dystonia and other symptoms. The syndrome is caused by mutations in the TIMM8A gene which codes for one of two human Tim8 isoforms, hTim8a. Kang et al. identified a new role of the human Tim8a protein in the assembly of respiratory chain complex IV, and demonstrated that this function is highly specific for neuronal cells. As a consequence of the complex IV assembly defect, the cells lacking Tim8a overproduce reactive oxygen species and are sensitized to apoptosis. Vitamin E was able to mitigate some of the secondary defects caused by the TIMM8A mutations although it did not repair the primary COX deficiency.

Overall, the data reveal an exciting new insight into the poorly defined patho-mechanism of the severe disease and shed light on a phenomenon of tissue specificity and mitochondria related disease.

**Decision letter after peer review:**

Thank you for submitting your article "Neuronal-specific function of hTim8a in Complex IV assembly provides insight into Mohr-Tranebjærg syndrome" for consideration by *eLife*. Your article has been reviewed by three peer reviewers, and the evaluation has been overseen by a Reviewing Editor and Vivek Malhotra as the Senior Editor. The following individual involved in review of your submission has agreed to reveal their identity: Johannes M Herrmann (Reviewer #3).

The reviewers have discussed the reviews with one another and the Reviewing Editor has drafted this decision to help you prepare a revised submission.

The reviewers are quite excited about the findings. The results are sound, of high quality and interesting for a broad audience. However, several major points need to be addressed, experimentally and by text clarifications, to improve the manuscript.

1) The authors claim that hTim8 has different substrates in different cell types. However, the approaches used here are not exactly identical (for example, hTim8a is deleted in HEK293 cells whereas about 40% of SH-SYS5 cells still contain WT alleles). Also, the isoforms are not comprehensively investigated (for example, it would seem more appropriate to perform the study in Tim8b KO cells to investigate the role of complex IV in the apoptosis response). The study would be stronger if the authors could unify some of the critical experimental schemes (potentially probing also for direct interactions to confirm substrate specificity). Also, the remaining differences in approaches should be acknowledged in the text.

2) Import assays analyzing the assembly of complex IV subunits in the mutants will strengthen the message of the manuscript. They would help to answer an important question which COX factors interact with hTim8 and whether Tim8-Tim13 is critical for the import of these proteins or whether it is part of a COX assembly complex and hence forms persistent contacts to COX subunits.

3) The authors made an impressive effort to study hTim8a interactors by mass spectrometry. However, the authors only highlighted very selective components in these graphs (e.g. Figure 3C) to support their conclusions (e.g. on the hTim8a-COX interaction). Follow-up experiments to identify direct substrates of the human Tim8-Tim13 complex and to confirm the proteomic analyses are missing. What are the direct interaction partners of hTim8 in SH-SYS5 cells?

4) In addition to complex IV defects, the authors also refer to a defect in complex I (Figure 2D). In addition, immunoblotting of several mitochondrial and nuclear encoded complex IV and complex I subunits in the hTim8a mutant cell lines will complete the dataset. It would be interesting to assess oxygen consumption and complex activities in Tim8a-deficient cells.

5) In relation to the points 3 and 4 above, the authors propose a series of events in the following order: TIMM8a -> COX assembly defect -> ROS production -> Apoptosis -> pathological consequences. Presumably the human Tim8-Tim13 complex serves a number of different clients, such as carriers, TIM22 components and COX constituents, or complex I and their absence causes a more pleiotropic defect. How confident can we be that it is complex IV disassembly in Tim8a-deficient SH-SY5Y cells that is responsible for priming the cells to apoptosis?

Specific comments (discussed and unified by the reviewers):

- In most cases, to quantify apoptosis, the authors measure PS exposure using the Annexin V assay and the results are expressed as 'relative luminescence unit'. Although this is useful, it would be more informative to include the proportion of cells which die in these assays.

- It is stated in the text that hTim8bKO HEK293 cells show reduced levels of Cox16 in Figure 2B, but from the data shown, it looks more like Cox16 levels are enhanced. This point should be clarified.

- The hTim8 proteomic data set is missing. The text refers to Table 3, however, this table contains the metabolomics data. This needs to be fixed.

---

## [Author Response]

The reviewers are quite excited about the findings. The results are sound, of high quality and interesting for a broad audience. However, several major points need to be addressed, experimentally and by text clarifications, to improve the manuscript.1) The authors claim that hTim8 has different substrates in different cell types. However, the approaches used here are not exactly identical (for example, hTim8a is deleted in HEK293 cells whereas about 40% of SH-SYS5 cells still contain WT alleles).

We don't suggest that hTim8a has different "substrates" in HEK293 versus SH-SY5Y cells, but rather suggest that the protein is functionally more active in SH-SY5Y cells consistent with the tissue specific phenotype seen in patients with mutated hTim8a. Is it important to note that Complex IV related pathologies also have a strong tissue specificity and we have addressed this in the revised Discussion. We have also included significant new data to show the cell specific function of the hTim8a and hTim8b isoforms. See comment 1-2 below.

To clarify the comment on the presence of WT alleles in the SH-SY5Y hTim8a cell line (hTim8a^MUT SH^): we are not using a mixed population of cells – i.e. 40% of these cells do *not* contain the wild type allele as indicated in this comment, rather hTim8a^MUT SH^ is a clonal cell line that contains one modified allele (c.[54_55insT]) and one wild type allele (Figure 1—figure supplement 1). Immunoblotting and mitochondrial proteomics (Figure 1—figure supplement 2 and Figure 2) confirm the absence of any detectable hTim8a protein. Given the presence of a wild-type allele, but no protein we conclude as indicated in the text "It has been reported that the expression of hTim8a can be altered by skewed X-chromosome inactivation (Plenge et al., 1999), thus we hypothesise that the observed wild-type allele of TIMM8A is located on an inactive X-chromosome in these cells. As this cell line is not a complete knockout, we refer to it as hTim8a^MUT SH^."

Also, the isoforms are not comprehensively investigated (for example, it would seem more appropriate to perform the study in Tim8b KO cells to investigate the role of complex IV in the apoptosis response). The study would be stronger if the authors could unify some of the critical experimental schemes (potentially probing also for direct interactions to confirm substrate specificity). Also, the remaining differences in approaches should be acknowledged in the text.

We include the following new data, which unifies the approaches employed for Tim8a and Tim8b and has allowed us to show that hTim8a has a more prominent function in SH-SY5Y cells and hTim8b a more prominent function in HEK293 cells.

1) We created *TIMM8B* CRISPR/Cas9 KO cell line in SH-SY5Y cells (Tim8b^KO-SH^). The study now consists of 4 cell lines (hTim8a^KO^ and hTim8b^KO^ (HEK293 background) and hTim8a^MUT SH^ and hTim8b^KO SH^ (SH-SY5Y background). We have performed the following analysis on all four cells lines, which has allowed us to dissect the cell specific function of the Tim8 isoforms, but also uncover a tissue specific role of this protein in Complex IV biology.

2) Mitochondrial proteomics (Figure 2) and BN-PAGE analysis (Figure 6) showing that both hTim8a and hTim8b influence the levels of Complex IV subunits and assembly factors in a cell specific manner.

3) Comprehensive cell and mitochondrial health measurements in Figure 4, including cell viability, mitochondrial membrane potential, oxygen consumption rates, and respiratory chain complex activity. This data highlights a more prominent role of hTim8b in HEK293 cells, while hTim8a has a critical role for SH-SY5Y health.

4) Import of key substrates including (Tim22, Tim23, GC1) (Figure 1) showing that the import of these substrates is not dependent on hTim8a/hTim8b.

5) Oxidative stress (Figure 9A) and apoptosis sensitivity assays (Figure 9B) showing the hTim8a neurons are more prone to both of these measures than hTim8b.

2) Import assays analyzing the assembly of complex IV subunits in the mutants will strengthen the message of the manuscript. They would help to answer an important question which COX factors interact with hTim8 and whether Tim8-Tim13 is critical for the import of these proteins or whether it is part of a COX assembly complex and hence forms persistent contacts to COX subunits.

in vitro imports of Complex IV subunits (COX4, COX6A, COX8A) and Complex I (NDUFV3) (Figure 6—figure supplement 1) did not reveal import defects or assembly intermediates, suggesting that hTim8a/b-hTim13 complex is not involved in the import of these proteins into mitochondria. Our data collectively suggests that hTim8a is involved in the general assembly of Complex IV possibly as an assembly factor acting within the network of assembly factors. New Figure 7G shows a hTim8^FLAG^ immunoprecipitation from cells grown on glucose (new experiment) versus galactose (in original manuscript). Interestingly, we show that the most enriched partners of hTim8a in situations of low respiratory demand (glucose) are hTim8b and hTim13, in addition to Coa4, Coa7, cyt *c* and COX6B1. Under high respiratory demand, hTim13 becomes far less enriched and we see the addition of Coa6 and COX41l to this interaction network. This suggest a rearrangement of the hTim8a/b/13 complex and based on crosslinking of hTim8a to Cox17 (Figure 7F) we conclude that hTim8a is acting in the modular assembly pathway of Complex IV either as an assembly factor or regulator of other assembly factors. Unravelling the precise molecular function of Tim8 is beyond the scope of this paper and presents an exciting avenue of future research.

3) The authors made an impressive effort to study hTim8a interactors by mass spectrometry. However, the authors only highlighted very selective components in these graphs (e.g. Figure 3C) to support their conclusions (e.g. on the hTim8a-COX interaction). Follow-up experiments to identify direct substrates of the human Tim8-Tim13 complex and to confirm the proteomic analyses are missing. What are the direct interaction partners of hTim8 in SH-SYS5 cells?

We apologise for this and now adjusted new 7G (old Figure 3C) to include labelling of additional enriched proteins. We have also included Figure 7—source data 1 showing a ranked list of all enriched proteins in the IP. To address the stable interacting partners, we performed the experiment described above in point 2, x-linking and immunoprecipitation of hTim8a^FLAG^ from mitochondria grown on glucose as a carbon source. This data reveals that one of the most stable interacting partners is indeed hTim13, however this interaction is compromised under conditions of high respiration. This exciting result indicates the hTim8a/b/hTim13 complex is not a static entity but can rearrange to accommodate unique functions.

As indicated in the Discussion our data suggests that the Tim8a/Tim8b/Tim13 network is involved in a number of aspects of mitochondria biology. We see interactions among the chaperones themselves, but also with proteins involved in Complex IV biology and other proteins involved in mitochondrial quality control, including pGAM5, OPA1, Yme1L, CLPB and the Prohibitins. We have not addressed these additional interactions in the context of this study, but again this will likely represent an exciting avenue for future research.

4) In addition to complex IV defects, the authors also refer to a defect in complex I (Figure 2D). In addition, immunoblotting of several mitochondrial and nuclear encoded complex IV and complex I subunits in the hTim8a mutant cell lines will complete the dataset. It would be interesting to assess oxygen consumption and complex activities in Tim8a-deficient cells.

We now include new Figure 6—figure supplement 1A, which include immunoblots and quantifications for Complex I and Complex IV subunits in mitochondrial lacking hTim8b or hTim8a. We also include Figure 6A, which addresses the reviewers comment more directly and shows the relative abundances of all Complex I and Complex IV subunits across the four cell lines used in this study (hTim8a^KO^, hTim8^MUT SH^, hTim8b^KO^ and hTim8b^KO SH^). Through this analysis we can see that hTim8b^KO^ HEK293, hTim8a^MUT^ SHSY5Y and hTim8b^KO^ SHSY5Y cells have decreased abundance in Complex IV subunits or assembly factors compared to control cells. hTim8a^KO^ HEK293 do not share this profile, reflecting the tissue-expression and/or function of hTim8a. Despite the downward trend of Complex I (Figure 6A, middle panel), few individual subunits are depleted significantly, and we believe this reflects a co-dependency of Complexes I and IV for stability in respiratory supercomplexes.

Oxygen consumption rate measurements (new Figure 4C, D and E) indicate no major respiration defect in HEK293 cells lacking hTim8a or hTim8b, however a minor reduction in maximal OCR of hTim8b^KO^ HEK293 following the addition of FCCP. Importantly, SH-SY5Y cells lacking hTim8a are severely impaired in both basal and maximal OCR, which correlates well the individual respiratory chain complex activities in these cells (new Figure 4F showing defects in complex I and IV in SH-SY5Y cells lacking either hTim8a or hTim8b).

5) In relation to the points 3 and 4 above, the authors propose a series of events in the following order: TIMM8a -> COX assembly defect -> ROS production -> Apoptosis -> pathological consequences. Presumably the human Tim8-Tim13 complex serves a number of different clients, such as carriers, TIM22 components and COX constituents, or complex I and their absence causes a more pleiotropic defect. How confident can we be that it is complex IV disassembly in Tim8a-deficient SH-SY5Y cells that is responsible for priming the cells to apoptosis?

Given that all four cell lines created in this study had reduced levels in the COX17 protein we wondered if a COX17-dependent process was at the heart of the cell death phenomenon in cells lacking hTim8a. To dissect this, we have included new Figure 9. We show that knockdown of COX17 results in the same cellular outcomes as the depletion of hTim8a (upregulation of cytochrome *c*, decrease in mitochondrial membrane potential, enhanced sensitivity to apoptosis, elevated oxidative stress), suggesting a link. In a more direct approach, we then assessed if CMV driven re-expression of COX17 in hTim8^MUT^ SH-SY5Y cells could restore some of the cellular phenotypes observed in these cells. As COX17 is an established substrate of Mia40 we knew the protein should have no issues reaching the mitochondria. Indeed, this approach complemented many of the phenotypes observed in cells lacking hTim8a, including a reduction in ROS levels and reduction in cell sensitivity to death. We believe this data suggests that the Complex IV assembly defect is driving the pathology associated with MTS, however we do agree with the reviewer in that we cannot dismiss the possibility that some other protein and/or substrate of hTim8a is playing a role. Indeed, we indicate this in the Discussion.

Specific comments (discussed and unified by the reviewers):- In most cases, to quantify apoptosis, the authors measure PS exposure using the Annexin V assay and the results are expressed as 'relative luminescence unit'. Although this is useful, it would be more informative to include the proportion of cells which die in these assays.

The apoptosis kit utilised in our studies provides information on the rate of apoptosis (detecting the amount of PS localised to the outer cellular membrane in real time), not an absolute number of cells undergoing death. To address the proportion of total cells in the population that are apoptotic, we have stained control and hTim8a^KO^ cells using a fluorescence-labelled-Annexin V and sort a total of 20,000 individual cells each using the fluorescence-based cell sorter to quantify the exact% of cells that are apoptotic, and show this in new Figure 3—figure supplement 1B.

- It is stated in the text that hTim8bKO HEK293 cells show reduced levels of Cox16 in Figure 2B, but from the data shown, it looks more like Cox16 levels are enhanced. This point should be clarified.

We apologise for this typographical error and have corrected this sentence.

- The hTim8 proteomic data set is missing. The text refers to Table 3, however, this table contains the metabolomics data. This needs to be fixed.

This has now been fixed and appropriate source files are associated with each figure.